

# Detection and characterization of precipitation extremes and geohydrological hazards over a transboundary Alpine area based on different methods and climate datasets

Alice Crespi[1*], Katharina Enigl[2,3*], Sebastian Lehner,[2,3] Klaus Haslinger[2] and Massimiliano Pittore[1]

[1] Center for Climate Change and Transformation, Eurac Research, Bolzano, Italy
[2] Department Climate – Impact – Research, GeoSphere Austria, Vienna, Austria
[3] Department of Meteorology and Geophysics, University of Vienna, Vienna, Austria

[*] These authors contributed equally to this work.

*Correspondence to*: Alice Crespi, alice.crespi@eurac.edu

**Abstract.** Extreme hydrometeorological events are increasingly raising concern in central Europe, particularly in the European Alps, where they pose significant threats to ecological and socio-economic systems. To support authorities to improve risk reduction and climate change adaptation efforts it is crucial to understand upon which conditions the available meteorological data allow for the detection of meteorological extremes able to trigger hazardous events in a given area of interest. Considering precipitation as a key triggering factor for such hazards, this study explores different approaches for the identification of extreme precipitation events and the assessment of their link to geohydrological processes (i.e., landslides, debris flows, floods) observed in a transboundary Alpine region between Austria and Italy from 2003 to 2020. Three definitions of extremes based on regional and local-scale statistics were applied to the daily precipitation grids from four meteorological datasets and the events identified by each combination of datasets and statistical approaches were then compared with hazard occurrences both spatially and temporally. Results show that daily precipitation fields identified as extreme by local-scale statistics, i.e., considering local intensities, report a greater spatial and temporal match with observed hazards. High-resolution observation products, especially if in situ observations are combined with radar data, offer a more detailed and reliable representation of precipitation intensities and relation with hazards. For all methods, the coarser-resolution reanalysis ERA5-Land shows the lowest performance in explaining hazard occurrences, mostly due to the gap between the spatial scale resolved by the data and the one relevant for geohydrological processes. The precipitation statistics and the fields of extreme events identified for the considered region are provided as a reference for further studies. The outcomes of this work can provide methodological recommendations for supporting the understanding and modelling of transboundary risks related to precipitation extremes triggering geohydrological processes in the Alpine regions.

## 1 Introduction

Weather extremes such as heavy rainfall events can trigger multiple hazardous processes leading to adverse consequences on ecosystems and human activities. Intense precipitation can cascade into geohydrological hazards, such as floods and gravitational mass movements like debris flows, landslides and rockfalls. These phenomena could in turn cause loss of life and damage buildings, transport infrastructure, agricultural production, and natural capital also with long-term economic effects (Bouwer, 2019). The costs of disaster response and recovery can further strain economies, emphasizing the need for understanding and preparing for extreme events to enhance resilience and reduce vulnerability. In 2023 in Europe, for instance, extreme hydrometeorological events incurred more than 20 fatalities and 15 billion Euro of losses (Munich RE, 2024).

The Alpine region, characterized by a complex topography, is particularly prone to geohydrological hazards mainly triggered by intense precipitation episodes, including heavy snowfall, whose effects can transcend national borders and affect multiple countries simultaneously (Steger et al., 2023; Stoffel et al., 2014). Precipitation intensity and extreme occurrence are generally assumed as direct proxies for potential geohydrological phenomena, as well as for the related risks, although some of these processes can also be caused by complex cascading or compound processes or be influenced by multiple predisposing factors (Gonzalez et al., 2024; Steger et al., 2023; Zscheischler et al., 2020).With the expected intensification of precipitation extremes in the Alpine region under future warming scenarios, the potential for hazardous processes could also increase, thus, posing additional challenges in an already vulnerable territory (Kotlarski et al., 2023; Rajczak and Schär, 2017; Wilhelm et al., 2022).



45 Understanding hazardous precipitation extremes in the Alpine region is, therefore, increasingly important to support effective risk management, including early-warning systems, and climate-change adaptation strategies for mitigating adverse consequences on natural systems and human communities. In this work we aim at: a) exploring sound and systematic approaches to detect and characterize extreme precipitation events from various gridded meteorological datasets featuring different spatial resolutions and b) investigating the relation between detected extreme events and observed hazards.

50 Various definitions of "extreme" can generally be adopted depending on different characteristics of the phenomenon in question and the specific application (McPhillips et al., 2018; Stephenson, 2008). Hereby, we consider extremes as environmental conditions whose rate of anomalousness in terms of intensity and areal extent compared to a statistical reference distribution exceeds a given threshold. In the case of a univariate empirical variable, this threshold could be represented by a specific percentile (e.g., 99[th]) of a probability distribution derived from available data over a reference period (Seneviratne et
55 al., 2021; McPhillips et al., 2018) or just by a fixed magnitude (Barlow et al., 2019). Understanding the sensitivity of precipitation extreme definitions with respect to the temporal and spatial dimensions of the underlying processes and area of interest is essential to perform a robust characterization of past rainfall events (Pinto et al., 2013; Raj et al., 2021; Ramos et al., 2014). Moreover, the identification accuracy for precipitation events also depends on the meteorological dataset used. Gridded precipitation products provide a continuous spatial description of precipitation fields with spatial details depending
60 on the type of data source used for their development. High-resolution, e.g., km-scale, datasets derived by interpolating station observations or integrating in situ and radar measurements are often employed to derive a detailed local representation of precipitation patterns. In the absence of such datasets, as in case of wide study areas crossing multiple countries for which a consistent high-resolution trans-border observation product may not exist, reanalysis data, i.e., simulations of atmospheric models constrained by observations, are valuable alternatives, even though they generally offer a coarser spatial resolution.
65 The ability of regional gridded observations and reanalysis data to represent daily precipitation extremes has been evaluated extensively (e.g., Bandhauer et al., 2022; Hu and Franzke, 2020; Reder et al., 2022). The performance of these datasets is influenced not only by their resolution but also by the different methods used to produce the spatial fields (Alexander et al., 2020; Hu and Franzke, 2020).

 Besides the representation of precipitation intensities, understanding to which extent precipitation events identified as extreme
70 correspond to hazards is a crucial step for developing impact-related detection frameworks supporting risk-oriented studies. There exists a knowledge gap in evaluating the skill of gridded meteorological data and statistical definitions of extremes to explain hazard records, with only a limited number of studies addressing this issue. For instance, Insua-Costa et al. (2021) identified and ranked past precipitation events in the Western Mediterranean area by applying a spatially varying daily precipitation threshold to a 5.5-km precipitation dataset (Soci et al., 2016) and linked them to flooding occurrences reported
75 in international disaster databases (Llasat et al., 2013). Liu et al. (2020) employed a percentile-based method to detect extreme precipitation events in the mainland of China and connected those to economic losses caused by floods to derive a disaster-triggering threshold of extreme precipitation. Wood et al. (2024) conducted a systematic evaluation of precipitation, temperature, and snowfall metrics across four climate reanalysis datasets, assessing their ability to represent the frequency and intensity of extreme hydrological events in Switzerland. However, a systematic inter-comparison of statistical definitions and
80 available precipitation datasets for detecting hazard-related precipitation events is lacking in existing literature.

 This study addresses this gap by focusing on the identification and assessment of precipitation extremes and their connection to geohydrological hazards over a transboundary Alpine area between Italy and Austria, employing a comparative approach that leverages various methods for extreme definition and different meteorological datasets. The presented analyses and findings aim at providing actionable information for risk practitioners and civil protection authorities to better understand how





and to which extent available datasets can be used for assessing impacts and risks related to precipitation extremes and for supporting local risk mitigation and climate-change adaptation measures (European Environment Agency, 2024).

## 2. Material and Methods

### 2.1 Study area

The study area is a transboundary region in the eastern European Alps including South Tyrol, the northernmost province of

Italy, as well as East Tyrol and Carinthia in the southern part of Austria (Figure 1). The region extends over ~ 19,000 km² mostly on the southern Alpine ridge and it is dominated by a mountainous landscape with strong altitude gradients. South Tyrol extends over 7,400 km² with elevations ranging from 200 to 3,900 m above sea level (a.s.l.) and about 85 % of the territory is located above 1,000 m a.s.l.. In East Tyrol and Carinthia, the elevation ranges from 400 to 3,800 m a.s.l. with about 75 % of the territory (~ 11,500 km²) above 1,000 m a.s.l.. The climate is influenced by the humid airflows from the Atlantic

North-West, including air masses from the continental east and air masses advected from the south, with occasional so-called Vb cyclone patterns induced through Genoa-cyclogenesis (Hofstätter et al., 2016). Precipitation is primarily driven by mesoscale systems, with the highest precipitation amounts occurring in summer and secondarily in autumn. The complex topography induces local patterns, such as typical drier conditions in inner valleys due to the sheltering effects of surrounding mountains (Price, 2009).

South Tyrol is populated by 530,000 people (ASTAT, 2023), of which one fifth live in the provincial capital Bolzano. In East Tyrol and Carinthia, most of the population (approximately 610,000 people) live in the regional capitals Lienz (East Tyrol) and Klagenfurt (Carinthia). The transboundary area is a popular touristic destination. In 2023 about 36 million overnight stays were recorded in South Tyrol, 2.5 million in East Tyrol and 13 million in Carinthia (ASTAT, 2024; Statistik Austria, 2024). Tourism, together with agriculture and manufacturing are the most important economic sectors of this territory. Due to the

mountainous topography, main settlements and infrastructure are concentrated in the valleys making them vulnerable to natural hazards typical of the surrounding steep terrain, especially gravitational mass movements including landslides, rockfalls and avalanches in higher elevations (Pittore et al., 2023; Schlögel et al., 2020). Moreover, the region hosts critical infrastructures of high social and economic relevance connecting the two national areas, such as the corridors of the Trans-European Transport Network crossing the Brenner pass.

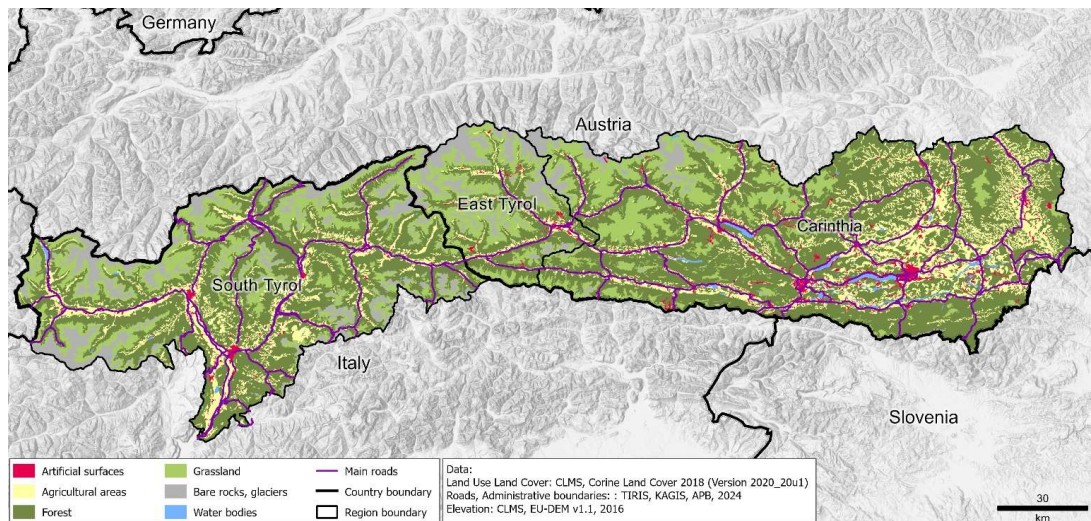

**Figure 1:** Map of the Italy-Austria cross-boundary study area depicting the main roads and land-use classes. Data sources are reported in the legend.



## 2.2 Data

### 2.2.1 Climate datasets

**SPARTACUS-TST**

To derive an observation dataset covering the transboundary Italy-Austria study area, two available regional gridded products available for each sub-region were merged. The Spatiotemporal Reanalysis Dataset for Climate in Austria (SPARTACUS, Hiebl and Frei, 2018), providing daily precipitation fields for Austria, covers East Tirol and Carinthia, while the regional dataset developed by Crespi et al. (2021) for Trentino-South Tyrol, hereafter TST, covers South Tyrol. SPARTACUS was
generated by interpolating quality checked data from irregularly distributed weather stations. It provides daily precipitation totals from 1961 onwards on a 1-km grid across Austria and close surroundings, i.e., the southern parts of Germany and South Tyrol in Italy. However, the low number of weather stations from South Tyrol (< 10) included in the SPARTACUS interpolation scheme limits the accuracy of reproduced precipitation patterns in this area (Figure S1). A preliminary comparison with rain gauge observations and TST precipitation fields across South Tyrol showed that the SPARTACUS
dataset underestimates precipitation intensities over the Italian area with some events being completely missed. For this reason, the TST dataset was used for the Italian part of the study region providing daily precipitation totals from 1980 onwards on a 250-m grid based on the records from about 80 rain gauges located in South Tyrol. The spatial combination of the two products was considered possible since these are generated by a similar two-step interpolation algorithm modelling the dependence of precipitation on orographic features. The TST dataset was thus bilinearly interpolated onto the 1-km grid of SPARTACUS
and replaced it over South Tyrol. The temporal dimension of TST was also realigned with SPARTACUS, which refers to precipitation totals from 06 UTC of the current day to 06 UTC of the following day, while precipitation totals in TST are from 08 UTC of the previous day to 08 UTC of the current day. Besides the temporal alignment and similar interpolation scheme, some minor spatial discontinuities in the merged fields can remain in the border area due to the different weather stations and the level of smoothing of orographic details used in the interpolation procedure of SPARTACUS and TST. Although these
discrepancies are expected to have a negligible impact on the event analysis, the adopted approaches were elaborated to minimize the effect on the results of potential discontinuities due to residual temporal shifts in the merged daily fields (see Sect. 2.3 for the methodology description). The merged dataset is hereafter referred to as SPARTACUS-TST.

**INCA**

The Integrated Nowcasting through Comprehensive Analysis (INCA) system offers high-resolution precipitation data on a 1-
km grid covering Austria and surrounding areas, including South Tyrol. It starts in 2003 and is updated every 15 minutes. INCA uses a combination of rain gauge data, weather radar estimates and detailed topographical information (Ghaemi et al., 2021; Haiden et al., 2011). The system integrates precipitation data from approximately 250 weather stations of the national network managed by Geosphere Austria and additional observations from the Austrian hydrographic service and surrounding areas. These observations are interpolated using an inverse distance weighting method and merged with radar data following
a specific correction procedure of radar estimates by means of the weather station records. In regions where radar coverage is insufficient, an elevation gradient is applied to the rain gauge data to enhance the overall accuracy. However, it is important to note that these corrections can sometimes result in very high scaling factors, which may create local artifacts in the final analysis (Ghaemi et al., 2021). INCA is considered a valuable tool for analysing spatial and temporal patterns of precipitation events in mountainous regions at a high resolution and is also suitable for transboundary assessments due to its spatial extent.
The 15-minute precipitation data of INCA were aggregated to daily sums, representing the precipitation totals from 00 UTC of the current day to 00 UTC of the next day.



### CERRA-Land

The Copernicus European Regional ReAnalysis (CERRA) is a regional reanalysis project developed by the European Centre for Medium-Range Weather Forecasts (ECMWF) for the pan-European region, featuring a horizontal resolution of 5.5 km (Ridal et al., 2024). CERRA was created using the HARMONIE-ALADIN limited-area numerical weather prediction and data assimilation system. CERRA reanalysis incorporates observational data, lateral boundary conditions from the ERA5 global reanalysis as prior estimates of the atmospheric state, and physiographic datasets describing the surface characteristics of the model. CERRA-Land provides surface and soil variables using the SURFEX land surface modelling platform and a daily total precipitation assimilation system. Specifically, daily precipitation fields are generated through an optimal interpolation scheme, combining an initial estimate derived from CERRA forecasted precipitation with daily rain gauge observations. Precipitation totals, available from September 1984 to May 2021 at a daily temporal resolution, represent the accumulated precipitation from 06 UTC of the previous day to 06 UTC of the current day. To ensure the temporal alignment with the other datasets, all daily records were shifted one day back. The primary advantage of CERRA-Land over other global reanalysis products lies in its higher horizontal resolution, which allows for a more detailed representation of topography and physiographic features.

### ERA5-Land

ERA5-Land is a replay of the land component of the global reanalysis ERA5 with a finer spatial resolution of 9 km and provides hourly fields of surface variables from 1950 to present. ERA5-Land does not assimilate observations, but it uses ERA5 atmospheric fields interpolated to the finer grid as input to control land model simulations. In addition, a lapse rate correction is applied to the forcing fields to account for the elevation difference between the 30-km grid of ERA5 and the target 9-km grid. Due to its spatial and temporal resolution and the temporal extent, ERA5-Land has been widely used for supporting impact-oriented applications and for assessing the state of the climate (Muñoz-Sabater et al., 2021). Daily precipitation totals are extracted from accumulated precipitation totals (including both liquid and solid precipitation) from 00 UTC of the current day to 00 UTC of the next day.



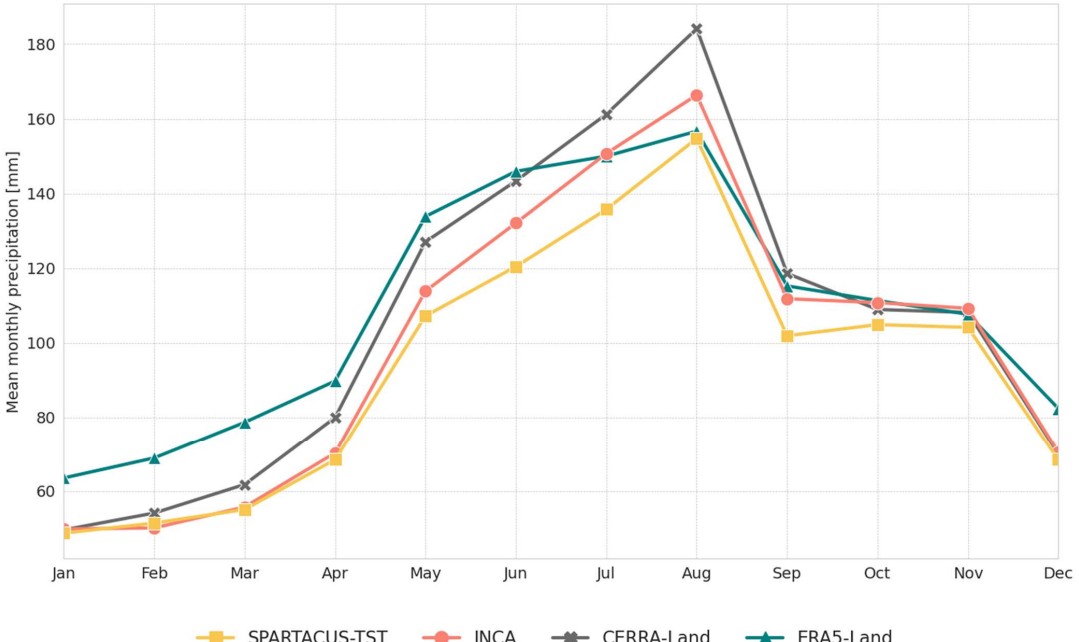

**Figure 2:** Mean monthly precipitation over 2003-2020 as spatial averages over the study area based on the different gridded datasets considered.

To enable the comparison, all following analyses were based on the congruent period 2003 to 2020 for all datasets, while each product was used in its native spatial resolution.

The four datasets reproduce a comparable annual distribution of the mean monthly precipitation totals over 2003-2020 as spatial averages over the entire domain (Figure 2), but with some clear differences in monthly magnitudes. CERRA-Land shows slightly higher precipitation totals during summer months, while ERA5-Land produces the wettest fields in all other months with up to + 40 % more precipitation in winter than the other products. The wet bias of ERA5-Land is coherent with the findings of previous inter-comparison studies (e.g., Monteiro and Morin, 2023; Dalla Torre et al., 2024). It is worth noting that the precipitation totals of SPARTACUS-TST are systematically lower than the values reported by the other datasets, especially in summer. Interestingly, the two reanalysis datasets show opposite characteristics for the mean annual cycle, with CERRA-Land depicting the strongest annual cycle and ERA5-Land the weakest.

The spatial distribution of mean precipitation totals in summer and winter half years (April to September and October to March, respectively), for SPARTACUS-TST and INCA are comparable, even though the orographic precipitation gradients are more distinct in INCA, especially during summer and over South Tyrol (Figure 3). This might be partly due to the orographic details considered in the production of INCA and to the inclusion of radar rainfall fields resolving finer spatial patterns than the observation-based grid and the coarser reanalyses. Orographic details are still evident in the 5.5-km grid of CERRA-Land, however, some grid cells show exceptionally high precipitation values, especially in summer, thus leading to the highest mean precipitation totals in summer months on a regional scale (Figure 3b). Due to the coarser resolution, the precipitation fields from ERA5-Land are the smoothest among all considered datasets.



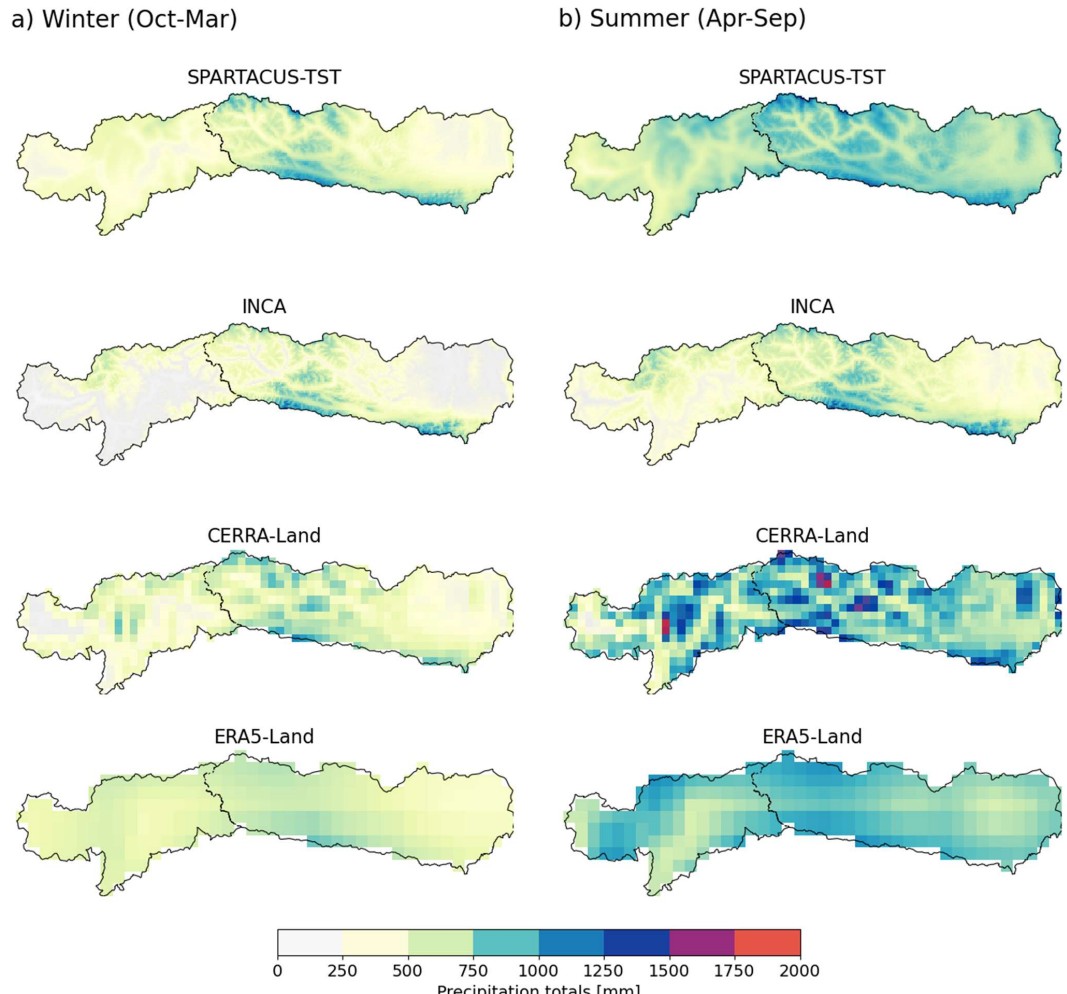

**Figure 3:** a) Winter half year (October to March) and b) summer half year (April to September) mean precipitation totals over 2003-2020 in the study area based on SPARTACUS-TST, INCA, CERRA-Land and ERA5-Land. Each dataset is shown in its native spatial resolution.

### 2.2.2 Hazard datasets

In this study, we use four different datasets that comprise recorded hazards in the study region. We focus on gravitational mass movements and floods, since these hazardous processes are closely linked to extreme precipitation events. It is important to note that different processes are included under the categories of floods and gravitational mass movements. For instance, the records for floods include fluvial sediment transports, urban flooding and overbank sedimentations, while gravitational mass movements encompass multiple movement types such as shallow landslides, debris avalanches, debris and mud flows, and rotational slides. It is important to note that although most hazards, especially floods, generally extend over an area, hazard records are often collected where actual impacts are observed and thus, they refer to specific locations. Due to the variety of processes covered by available records, their link to impacts, and their complex dependency on environmental conditions and conditioning factors, the use of daily precipitation extremes as triggering conditions provide only a first-order explanation of the occurrence of all observed hazards. Furthermore, in contrast to the case of meteorological data, there is currently no recognized standard for the collection and recording of hazard data, neither at cross-boundary level, nor at national level. To account for this additional bias the different types of geohydrological events have been treated as equivalent (no specific distinction is made) and only their recorded location and observation date have been considered.



### WLK

The Austrian Service for Torrent and Avalanche Control (WLV), established in 1884, has traditionally focused on managing torrents and avalanches primarily in Alpine regions. One of its key tools is the "Wildbach- und Lawinenkataster" (WLK), a digital geo-information management system introduced in 2017, which maintains comprehensive, long-term records of hazardous processes (Bundesministerium für Nachhaltigkeit und Tourismus, 2018). These records report geohydrological events pinpointed with exact locations. Additionally, most entries in the WLK include precise day of the event. In this study,

we use all documented events recorded between 2003 to 2020 for the region of Carinthia and East Tyrol featuring an exact event date.

### GEORIOS

   The Geological Survey of Austria, now operating under the name GeoSphere Austria, collects and maintains extensive records of gravitational mass movements within the GEORIOS database (Tilch et al., 2011). These records are derived from different

sources including literature, reports, and maps, leading to discrepancies in quality. Consequently, the precision and granularity of information related to location, scale, and volume feature notable variations across the database. Hence, a verification process is undertaken to rectify redundancies, and standardized methodologies are implemented to unify the data into a dataset with consistent quality (Tilch et al., 2011). The GEORIOS entries utilized in this study provide both event locations and days of occurrence spanning from 2003 to 2020, covering a range of gravitational processes.

**ED30**

   The "Event Documentation of the 30th Division of the Autonomous Province of Bolzano (ED30)" began in 1998 as a database for recording various incidents in the province of Bolzano, including natural disasters and human-made events ([https://pericoli-naturali.provincia.bz.it/it/archivio-report-pericoli-naturali](https://pericoli-naturali.provincia.bz.it/it/archivio-report-pericoli-naturali)). It helps authorities in analysing, planning, and responding to emergencies by providing detailed information on event dates, locations, impacts, and response actions. Data are sourced from

official reports, field assessments, eyewitness accounts and sensor data. This study focuses on the geohydrological events collected in this database from 2003 to 2020 for which the precise event location and date are available.

### IFFI

   The Italian Landslide Inventory (IFFI) serves as the national and official repository of landslide data. In collaboration with Regions and Self-Governing Provinces, the Italian National Institute for Environmental Protection and Research (ISPRA) has

overseen the implementation of IFFI. This inventory plays a crucial role as a fundamental knowledge resource for evaluating landslide hazards in River Basin Plans, conducting preliminary designs for slope instability and flood mitigation projects, and safeguarding infrastructure networks. Additionally, IFFI aids in the development of Civil Protection Emergency Plans. Currently, the IFFI database documents over 620,000 landslides across Italy. For this study, we focus on analysing gravitational mass movement events occurring in South Tyrol between 2003 and 2020, considering only those for which the day of

occurrence and point location are available.



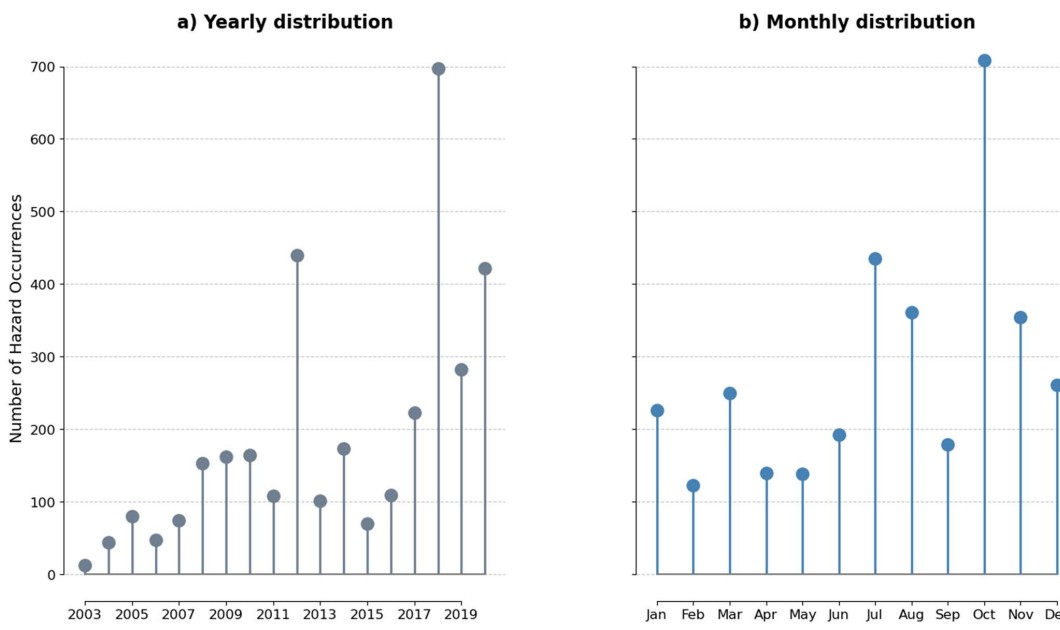

**Figure 4:** a) Annual and b) monthly distributions of documented geohydrological events in the study area over 2003-2020, sourced from the IFFI and ED30 databases for South Tyrol and the WLV and GEORIOS databases for Austria.

By merging all collected hazard events over the period 2003 to 2020 for the study region, a noticeable and statistically
significant increase (Mann-Kendall p-value < 0.01) in the number of records per year is observed (Figure 4a), although caution is advised in interpreting the temporal variability of the data over the 18-year period as a long-term trend. For instance, this increase might be partly related to a rise in the frequency of possibly damaging events, but also to the increasingly more systematic field collection of hazard observations. While in the past only hazard occurrences with a magnitude above a certain threshold were recorded, a greater number of hazard events started to be reported in more recent years (Heiser et al., 2019).

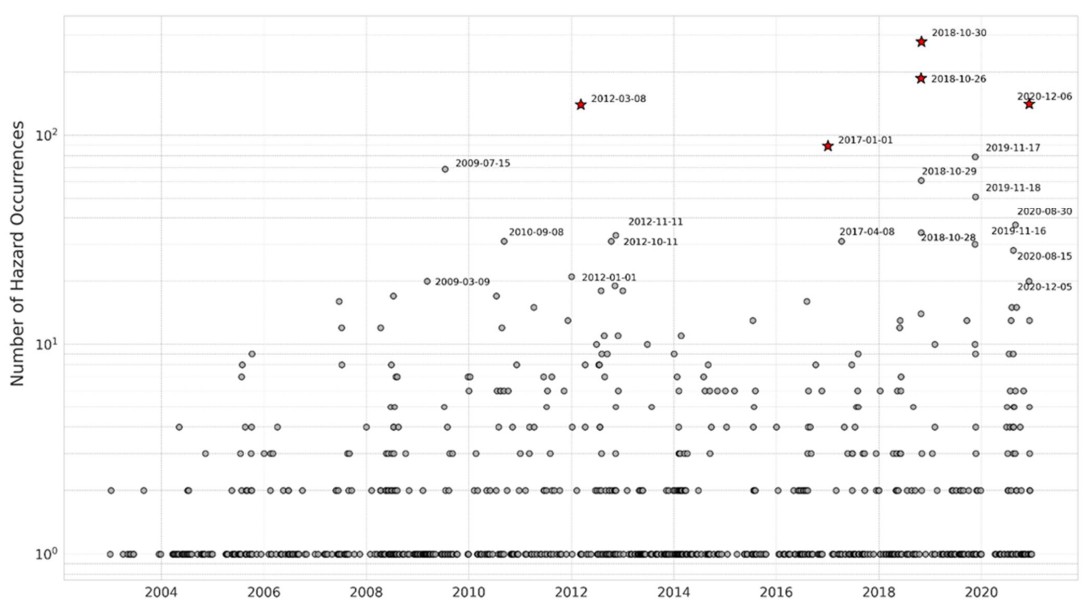






**Figure 5:** Distribution of daily recorded geohydrological hazards between 2003 and 2020. Dates with a star indicate the five episodes with the highest number of hazards recorded within the study area. The y axis is in logarithmic scale.

On a monthly level, the number of hazard records peaks in October and secondarily in summer months (Figure 4b). However, this distribution is strongly determined by few days with an exceptional number of recorded hazards (Figure 5). Between 1st

January 2003 and 31st December 2020, a total of 3,319 hazard events was recorded over 890 distinct days, resulting in a probability of observing a day with at least one recorded hazard of around 13.5 %, or almost one hazard event per week. The high frequency of days recording at least one geohydrological hazard underlines the orographic complexity of the study area determining very local processes. On the contrary, the dates with more hazard records can be associated with known meteorological extreme events. For instance, the highest number of hazard occurrences (279) was observed on 30th October

2018, during the Vaia storm. Similarly, a substantial number of geohydrological events was recorded in correspondence to intense storms from 16th to 18th November 2019 and on 5th and 6th December 2020, leading to 170 and 161 registered phenomena, respectively. These episodes are clearly reflected in the annual time series of hazard records indicating the highest number of occurrences in 2018 and 2020 (Figure 4a). It is worth noting that the second peak of the time series occurred in 2012, when multiple extreme meteorological events across the study area resulted in a high total of hazards recorded. The

relatively high probability of days with at least one geohydrological hazard occurrence in the study region and the concentration of records in correspondence to a few extreme meteorological episodes should be considered when analysing the link between hazard dates and extreme precipitation events (see Sect. 2.3). Moreover, the interpretation of this link should also consider that the daily time series of hazard records exhibits a significant autocorrelation until a lag of 4 days (Figure S2a).

## 2.3 Methodology

### 2.3.1 Event definition and detection

High-intensity precipitation events can be associated to large-scale phenomena affecting extended areas or be very localized in space and time, such as for convective processes. The level of intensity of an event can be described based on the absolute cumulated precipitation within a given area, or on its anomalousness relative to the precipitation conditions expected to occur in each given time of the year based on a reference period. Comparing event detection methods adopting different precipitation

intensity metrics allows for a more comprehensive evaluation of the ability to identify precipitation extremes coincident with hazard occurrences. The detection and characterization of past precipitation extremes in the study area are thus carried out by applying three different methodologies to the daily precipitation fields from four gridded products over the 2003-2020 period. For the purpose of the work, we assume that:

- the available precipitation data are a representative sample of the underlying distribution for the range of anomalies
we are interested in. We therefore analysed the empirical distributions and did not fit any theoretical extreme value distribution (e.g., Generalised Extreme Value or Generalised Pareto).

- the outcomes remain valid also in presence of non-stationarity, which was not explicitly considered.

- the daily cumulated precipitation is a sensible proxy for geohydrological hazards described by the collected records.

Each detection method applies a different, and complementary, score for measuring the regional intensity of precipitation

events, and based on the adopted definition, assigning a ranking to each day in the 2003-2020 period:

- In the first detection approach, hereafter "*areal mean*", dates in the 2003-2020 period are sorted based on their corresponding accumulated daily precipitation over all grid cells in the study region divided by the total number of grid cells (i.e., representing the average daily precipitation intensity per grid cell):

$$R_t = \frac{\sum_{i=1}^{N} r_{t,i}}{N} \qquad (1)$$



where $R_t$ is the sum at daily time step $t$ of the precipitation values over all $N$ grid cells of the study area divided by the total number of grid cells. The ranked values can be further filtered based on the percentile of the resulting distribution. This method is expected to mostly detect large-scale meteorological events leading to high precipitation amounts over a substantial portion of the study region.

-       The second approach, hereafter "*local p99*", consists in the calculation of the 99th percentile of precipitation values
over all grid cells for each day:

$$R_t^{99} = P_{99}(r_{t,1}, \ldots, r_{t,N}) \tag{2}$$

where $R_t^{99}$ is the 99th percentile at daily time step $t$ calculated over the precipitation values $r_t$ of all the $N$ grid cells in the study area. The dates from 2003 to 2020 are then ranked based on the daily 99th percentile value. The *local p99* approach is expected to capture precipitation events of high intensity but occurring on more localized sub-portions of
the study area.

-       The third approach, hereafter called "*anomaly*", measures the relative exceptionality of events based on the daily precipitation deviations from average conditions. While the two previous methods sort the daily fields based on their absolute precipitation values, the anomaly-based method assigns a higher ranking to dates with precipitation significantly exceeding the mean climatic conditions associated to that specific period of the year. More specifically,
by applying a similar procedure as described in previous studies (e.g., Insua-Costa et al., 2021; Ramos et al., 2014), daily precipitation time series for each grid cell $i$ are converted into standardized anomalies:

$$a_{t,i} = \frac{r_{t,i} - \mu_i}{\sigma_i} \tag{3}$$

where for each grid cell $i$, $a_{t,i}$ is the precipitation standardized anomaly at daily time step $t$, $r_{t,i}$ is the daily precipitation value, $\mu_i$ is the daily precipitation mean over all wet-day values (daily precipitation > 1 mm) in a 21-day window
centred on the corresponding calendar day and spanning the 2003-2020 period, while $\sigma$ is its standard deviation. For each day, the percentage of the grid cells with $a_{t,i} > 2$ and the mean value of these anomalies are computed. The product of the two represents the total magnitude of the event as a combination of its extent and anomaly. This score is then used to sort the dates.

The three methods are independently applied to all climate datasets, yielding three sorted lists of dates per dataset. Each list is used to derive and compare the topmost extreme precipitation days over 2003-2020 and to link them with the database of hazard occurrences. To ensure that subsequent dates belonging to the same event are excluded, any dates within a 5-day window in the top-event selection are filtered such that only the highest-ranking date is retained. The process continues by adding subsequent dates from the sorted full list until the target size of the date selection is achieved. (e.g., the top 5 % of the
sorted dates). Moreover, the daily statistics calculated for each dataset are also used for a preliminary assessment of the temporal variability of extreme precipitation characteristics in the study area over the period 2003 to 2020.

**2.3.2. Spatiotemporal comparison between identified precipitation extreme and hazard records**

To evaluate the causal relationship between identified extreme precipitation events and ensuing geohydrological hazards within the specified region, a comprehensive analysis is conducted by cross-referencing precipitation event occurrences with the
hazard records (date of occurrence and geographical location) collected from the national datasets (Sect. 2.2.2). The topmost extreme precipitation events over 2003-2020 to compare with hazards are defined by considering the top 5 % of the sorted precipitation dates obtained for each dataset-method combination. To align hazard records with the identified extreme precipitation events and make the results from different climate datasets comparable, the intersection is evaluated within a temporal window of ± 2 days around the precipitation event date in all analyses. This temporal buffer is expected to reduce
the effects on results of potential uncertainties in the reported hazard dates and possible shifts in the precipitation events recorded by different climate products. The window length is also justified by observing that the daily precipitation statistics



considered for event detection exhibit a significant cross correlation with the time series of daily hazard records until a lag of ± 3 days for all datasets (Figure S2b for SPARTACUS-TST only). This ensures a more robust alignment between detected precipitation events and associated hazard occurrences, thus facilitating the examination of their temporal correlation.

The cross-occurrence of extreme precipitation events and hazards in the region is first conducted regardless of the geographical location of the recorded hazards. The intersection is evaluated through contingency tables comparing the dates over 2003-2020 recording at least two geohydrological events over the region, hereafter called hazardous dates, with the 5-day windows centred on topmost precipitation extreme occurrences. The total number of days over 2003-2020 reporting at least two hazard records is 332, corresponding to ~ 5 % of the whole period and to 37 % of days with at least one hazard occurrence. A lower threshold

for filtering the series of hazard occurrences is intended to account for the relatively high frequency of days with only one hazard observation. This adjustment aims to prevent misrepresentation of the actual probability of an extreme precipitation event capturing hazardous dates (see Sect. 2.2.2). In case multiple hazardous dates are associated to the same 5-day precipitation window, they are counted only once. For each method and dataset, we evaluate the hit rate, which is here defined as the proportion of hazardous dates coincident with extreme precipitation events, which can be interpreted as the probability

for an hazardous episode of being properly captured by the top 5 % most extreme precipitation events. The contingency table is also used to estimate the conditional probability of recording an hazardous date given an extreme precipitation event and to assess the level of association between hazard occurrences and precipitation extremes. The statistical significance of the association is tested through a Pearson Chi-Square test considering a significance level of 0.05. The sensitivity to different thresholds for selecting hazardous dates for the intersection analysis as well as the variability of the hit rate per season, i.e.,

differentiating between winter and summer half years, are also investigated.

Since the evaluation of temporal association of hazard events with respect to precipitation extremes is conducted disregarding their specific geographical coordinates, it remains plausible that some hazard events may have occurred within the 5-day window surrounding an extreme precipitation event, but in locations where minimal or no precipitation was reported. To ascertain whether the recorded hazards align with instances of high precipitation as indicated by the various datasets, a detailed

examination of precipitation values in the spatial proximity to individual hazard events is conducted. In particular, for the identified extreme precipitation events registering hazards, the class of daily precipitation intensity associated to the location of hazard occurrences is investigated. To achieve this, the daily precipitation fields are categorized into six classes based on percentiles of the gridded values. The percentile ranges considered are: [0-0.1], [0.1-0.3], [0.3-0.5], [0.5-0.7], [0.7-0.9] and [0.9-1]. For each precipitation event, the number of recorded hazards within a 5-day centred window is counted and attributed

to the corresponding precipitation class. To avoid double-counting, the hazard records included in overlapping 5-day windows of subsequent events are considered only for the first event occurrence. Each hazard record is associated with the highest precipitation class of the four nearest grid cells. The total counts of hazards for each precipitation class for the top 5 % events for each dataset-method combination are subsequently derived and compared. In addition to this, we analyse the disparity in precipitation totals at the precise location of hazard occurrences across all dataset-method combinations. Specifically, for each

hazard record included in the 5-day windows centred on the top 5 % most extreme precipitation events given by the various dataset-method combinations, the precipitation total at the nearest grid cell on the day of occurrence of the precipitation event is extracted and used to analyse the relationship between hazard location and precipitation intensity. As for above, hazard records falling in overlapping 5-day windows are considered once and attributed to the first precipitation event.





# 3 Results

## 3.1 Analysis of precipitation extremes

### 3.1.1 Temporal variability of precipitation statistics

For all datasets included in the analysis, the temporal variability of daily precipitation statistics, specifically the areal mean, the local 99th percentile and the local maximum of daily precipitation values over the study region, is analysed. The time series of yearly averages of daily statistics are compared over the common period fully covered by the datasets (2003-2020) to identify ongoing tendencies in precipitation intensity and potential discrepancies among the climatic products.

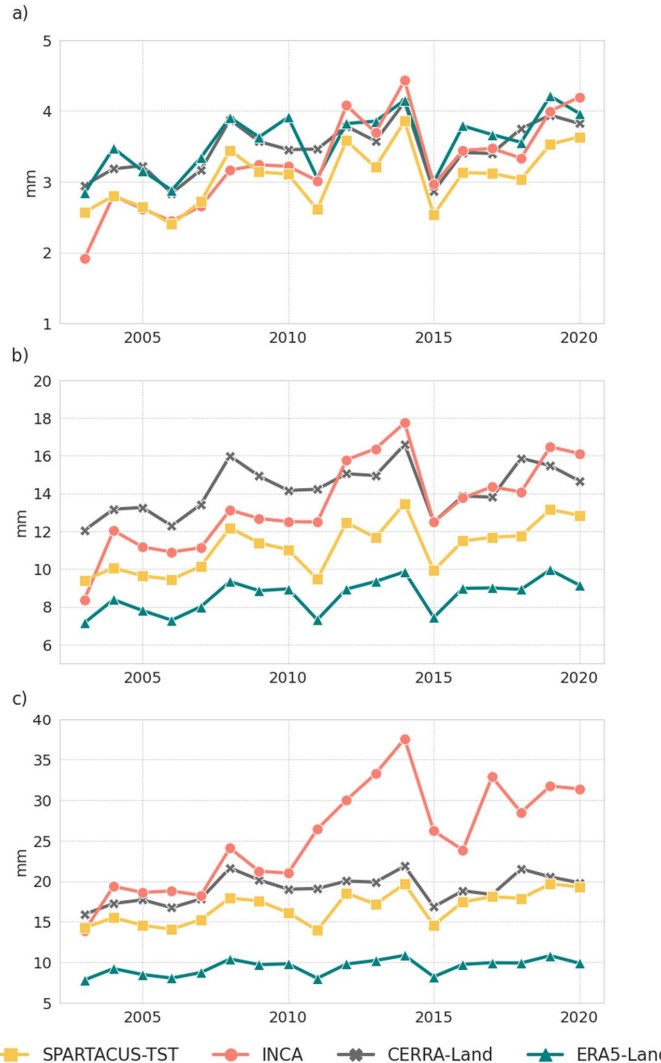

**Figure 6:** Annual time series 2003-2020 of the yearly averages of daily precipitation statistics: a) areal mean, b) local 99th percentile and c) local maxima over the study region.

Figure 6 shows a linear trend revealing a statistically significant ($p$-value $< 0.05$) increase over 2003-2020 of all precipitation statistics and in all climatic products, except for ERA5-Land and the local precipitation maxima (Figure 6c). It is interesting to note that while SPARTACUS-TST, CERRA-Land and ERA5-Land exhibit consistent patterns over the spanned period, INCA is characterized by more pronounced increases in all statistics. The increase in the areal mean (99th percentile) of daily



precipitation intensity with respect to the long-term average ranges from + 1.2 (1.0) % to + 1.5 (1.6) % per year in all datasets
except for INCA whose increase is about + 2.9 (2.6) % per year. The discrepancy of INCA is even more emphasized by the
annual averages of daily precipitation maxima with a sudden increment in the values after 2011. Consequently, while the
increase in local maxima for all other datasets is in the range + 1.0 % to + 1.5 % per year, it reaches + 3.9 % per year for INCA.
The greater positive trends of precipitation statistics derived from INCA are probably due to a discontinuity in the time series
after 2011 due to changes in the modelling frameworks. Although the increasing pattern is consistent among all datasets, the
length of the period considered for the analysis is too short (18 years) to conclude long-term increases in precipitation intensity
over the study region. To get a more comprehensive analysis of the temporal variability of precipitation intensity, the temporal
evolution of the annual statistics of precipitation intensity is extended over the maximum period (1985-2020) commonly
covered by the longer versions of SPARTACUS-TST, ERA5-Land and CERRA-Land datasets. All statistics (areal mean, local
$99^{th}$ percentile and local maxima) show increases over 1985-2020 in almost all cases with weaker signals for CERRA-Land.
In particular, trends are positive and statistically significant for SPARTACUS-TST and ERA5-Land for all statistics, while
CERRA-Land exhibits statistically significant increases in local precipitation maximum only. The trend in the annual average
of daily areal mean is ~ + 0.5 % per year for SPARTACUS-TST, with a 95 % confidence interval in the range + 0.1 % to +
0.9 %, and ~ + 0.4 % per year for ERA5-Land, with a 95 % confidence interval in the range + 0.1 % to + 0.7 % (Figure S3).
Similar trends and confidence intervals are exhibited by the other precipitation statistics calculated for SPARTACUS-TST and
ERA5-Land. The different temporal variability of CERRA-Land statics can be ascribed to the adopted assimilation system of
rain gauge observations, which, in case of changing observation networks over time, can introduces biases in the climatology
thus affecting the overall trend accuracy. ERA5-Land and SPARTACUS-TST are expected to be less influenced by the
temporal variability of in-situ data coverage since the first does not directly assimilate observations and the latter is based on
a stable set of historical observations. The increasing tendency exhibited over the extended period, with statistically significant
increases confirmed by SPARTACUS-TST and ERA5-Land for all statistics, is thus consistent with the recent temporal
evolution of precipitation intensity depicted over the shorter 2003-2020 period. It is also interesting to note that the interannual
variability of the annually aggregated statistics generally increases after 2000.

### 3.1.2 Characteristics of extreme precipitation events

The top 5 % of the most extreme precipitation events (corresponding to a total of 330 days) identified for each dataset-method
combination over 2003-2020 was first investigated to compare temporal patterns and assess seasonal dependencies.
The percentage of dates commonly detected based on different datasets ranges from 72 % to 88 % with the lowest fractions
for the local p99 and the highest ones for the areal mean. The greater dataset agreement in extreme precipitation dates detected
through the areal mean approach might be explained by the fact that areal-aggregated statistics are less sensitive to the spatial
details resolved by the datasets than statistics, i.e., the local $99^{th}$ percentile, calculated at the grid-cell level. For all methods,
the largest agreement in selected dates is obtained between SPARTACUS-TST and INCA and between SPARTACUS-TST
and CERRA-Land, while ERA5-Land generally shows the lowest portion of extreme precipitation dates in common with the
other datasets. Despite these differences, it is interesting to note that major precipitation events recorded in the study area in
recent years, such as the winter storm of December 2020[1], the Vaia storm in October 2018[2] and the severe precipitation
episodes in November 2019[3] and 2014[4], are detected by all datasets and methods, even if ranked differently. As an example,
the distribution of the top 5 % most extreme events detected by INCA based on the local p99 is shown in Figure 7 together
with the corresponding areal mean of daily precipitation values on the same dates.

---

[1] https://watchers.news/2020/12/07/major-winter-storm-snow-alps-december-2020/
[2] https://watchers.news/2018/11/04/severe-storms-italy-november-2018/
[3] https://weather.com/news/news/2019-11-18-austria-mudslides-flooding-avalanches-kill-one-person
[4] https://www.zamg.ac.at/cms/de/wetter/news/teils-extreme-regenmengen-in-osttirol-und-oberkaernten

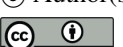



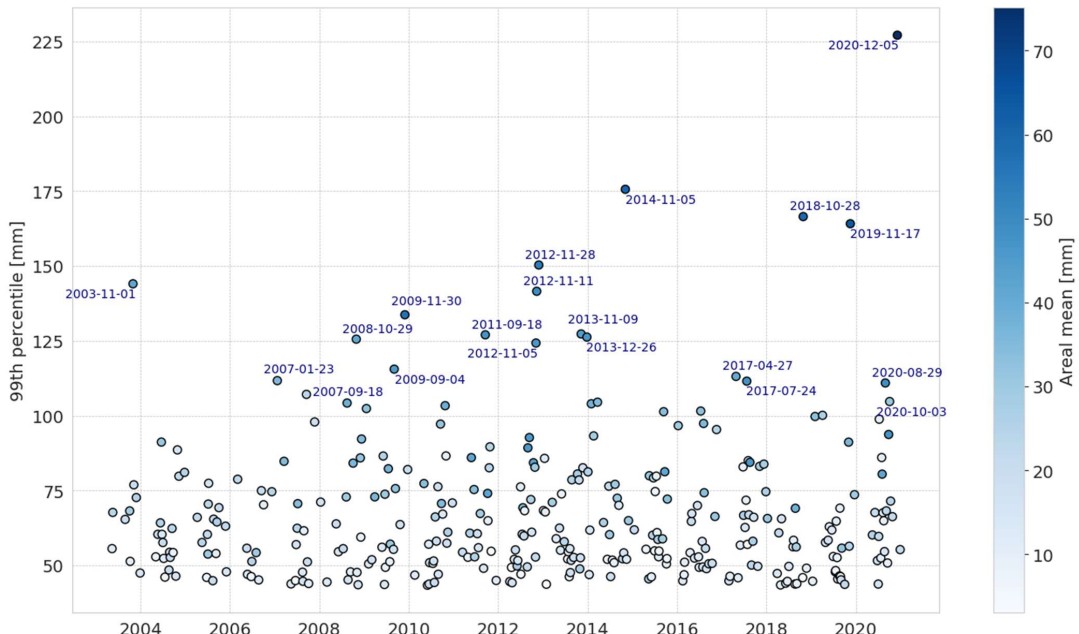

**Figure 7:** The topmost 5 % of precipitation events detected using INCA and the local p99 method in the study area. For each event the local 99th percentile (local p99 approach) used for ranking is shown on the y-axis. The colour of the point reports the areal mean precipitation (areal mean approach) for the same day, and the label refers to the date of the detected event.

The seasonal distribution of the top 5 % most extreme events is similar among all datasets considered (Table 1). Table 1 reports

the results for the three-month seasons instead of half-year seasons to provide a more detailed overview of the distribution of

precipitation extremes over the year. Most detected events by the areal mean and the local p99 occurred in summer and autumn.

This is generally consistent with the mean climate of the area experiencing the wettest conditions during summer and,

secondarily, autumn months (Figure 2). The seasonal distinction of events identified by the anomaly method is less

pronounced, which is expected since anomalies are standardized with the annual cycle, with a more substantial portion of

extreme episodes in spring compared to the other methods. Such spring events are probably not exceptional in terms of absolute

intensities but rather deviations from average conditions, which are usually drier than in summer and autumn.

| | SPARTACUS-TST | | | | INCA | | | | ERA5-Land | | | | CERRA-Land | | | |
|---|---|---|---|---|---|---|---|---|---|---|---|---|---|---|---|---|
| | MAM | JJA | SON | DJF | MAM | JJA | SON | DJF | MAM | JJA | SON | DJF | MAM | JJA | SON | DJF |
| **Areal mean** | 19 | 36 | 32 | 12 | 20 | 40 | 29 | 11 | 22 | 35 | 32 | 12 | 20 | 37 | 32 | 11 |
| **Local p99** | 18 | 38 | 28 | 15 | 15 | 44 | 29 | 12 | 19 | 37 | 32 | 13 | 17 | 44 | 29 | 10 |
| **Anomaly** | 27 | 38 | 22 | 14 | 25 | 39 | 21 | 15 | 27 | 33 | 21 | 18 | 27 | 36 | 22 | 15 |

**Table 1:** Relative distribution over seasons of the top 5 % most extreme precipitation events (330 events) over 2003-2020 detected by each dataset and method. For each dataset-method combination the relative portion (%) of total events falling in spring (March to May. MAM), 440    summer (June to August, JJA), autumn (September to November, SON) and winter (December to February, DJF) is reported.

While summer is mainly dominated by convective phenomena causing intense rainfall that is concentrated in space and time,

autumn is more prone to intense and persistent precipitation associated to low-pressure systems and affecting wider areas. The

highest portion of detected summer events is reported by INCA and CERRA-Land using the local p99, which might suggest

the greater ability of these datasets and method to represent locally intense precipitation events, particularly common during

the summer period. These findings are also reflected in the distribution of the 330 topmost events per year between 2003 and

2020 by distinguishing summer half year (from April to September) and winter half year (from October to March) (Figure S4).

In all cases, the 5 % topmost events occurred mainly in the summer half year. No dataset reports a visible trend in the annual

number of detected precipitation extremes except for INCA, which exhibits a significant increase in the annual counts of



summer events, in agreement with the temporal pattern of its extreme precipitation statistics as discussed in Sect. 3.1.1 (Figure
6). Notably, event occurrences and their temporal patterns are generally consistent among all datasets, regardless of the
detection method applied. For instance, all products show a similar peak of summer events in 2016 and a drop of winter events
in 2011 and 2015.

## 3.2 Spatiotemporal comparison of precipitation extremes with hazard records

### 3.2.1 Temporal association of precipitation and hazard occurrences

To obtain a preliminary assessment of how the extreme precipitation events at regional scale relate with observed hazard
occurrences, the series of daily counts of hazard records from 2003 to 2020 over the study area was correlated directly to the
daily precipitation statistics used in the detection methods, i.e., areal precipitation mean, local 99th percentile and event
magnitude (see Sect. 2.3.1). By calculating the cross-correlation between the daily counts of observed hazards and the daily
precipitation statistics, in all cases the correlation is the highest at lag + 1 day and remains significant until lag ± 3 days (Figure
S2b). This finding supports our choice of considering a buffer for evaluating the association of hazard records to a specific
precipitation event. However, the correlation coefficient with daily hazard records remains relatively low and varies little
among the precipitation statistics considered. By considering the lag + 1 day, the correlation between precipitation statistics
and hazards is ~ 0.16 for ERA5-Land and up to 0.22 for daily precipitation magnitude from SPARTACUS-TST (Table S1).
Correlation slightly increases if daily series are filtered by retaining only dates with at least two hazard records. It is interesting
to note that correlation with hazards is generally higher for precipitation statistics from SPARTACUS-TST and INCA.

The hit rate, defined as the proportion of the 332 dates with at least two recorded hazards (hazardous dates) coincident with
the top 5 % extreme precipitation events of each dataset-method combination, is the highest, on a yearly basis, for the local
p99 method for all climate datasets, even though it remains below 50 %. The greatest hit rate is reached by INCA (47.6 %)
and, secondarily, by SPARTACUS-TST (46.1 %), while the lowest values are reported by ERA5-Land regardless of the
method considered for detecting precipitation extremes (Table 2). In all cases, the level of association between extreme
precipitation occurrences and hazard episodes turns out to be significant (p-value < 0.05) with a conditional probability of
recording an hazardous date given the extreme precipitation event ranging between 31 % (ERA5-Land) and 39 % (INCA),
which is about 7 to 9-fold higher than the expected probability assuming hazard occurrence being independent of the happening
of precipitation extreme.

|  | Areal mean | | | Local p99 | | | Anomaly | | |
|---|---|---|---|---|---|---|---|---|---|
|  | Year | Summer | Winter | Year | Summer | Winter | Year | Summer | Winter |
| **SPARTACUS-TST** | 43.2 | 43.6 | 42.4 | 46.1 | 48.8 | 41.7 | 44.5 | 47.6 | 39.4 |
| **INCA** | 43.9 | 45.5 | 41.2 | 47.6 | 50.9 | 42.2 | 44.8 | 48.5 | 38.8 |
| **CERRA-Land** | 42.2 | 44.2 | 39.0 | 42.8 | 45.2 | 38.6 | 40.4 | 41.8 | 38.2 |
| **ERA5-Land** | 38.9 | 37.4 | 41.6 | 41.4 | 42.7 | 39.2 | 36.7 | 36.0 | 37.9 |

**Table 2:** Hit rate (%) as the portion of hazardous dates occurring in coincidence of an extreme precipitation event (within a 5-day window
centred on the extreme precipitation days) for each combination of datasets and methods and by considering the whole year, summer (April
to September) and winter (October to March) half years. The top 5 % most extreme precipitation dates over 2003-2020 and the hazard dates
with at least two hazard records are considered.

The same findings are obtained if a smaller sample of the hazard series is considered, e.g., by selecting only dates with at least
five hazard records, which correspond to ~ 1.7 % of the full series (117 dates). The hit rates are the highest for the local p99
for all datasets and greatest for INCA and CERRA-Land while the lowest correspondence with hazards is shown by ERA5-
Land.



By analysing the coincident occurrences of hazards and precipitation events in summer and winter half years, no specific differences among datasets and methods stand out (Table 2). The local p99 is confirmed to be the method providing the highest

hit rates for summer events in all cases, especially for INCA, while in winter slightly higher hit rates are reached by the areal mean for SPARTACUS-TST and ERA5-Land. The score values are generally higher in the summer half year, when extreme precipitation events occur more often than in the winter period. The only exception is exhibited by ERA5-Land whose hit rates are slightly higher for winter for both the areal mean and the anomaly approaches.

### 3.2.2 Spatial association of precipitation and hazard occurrences

Figure 8 shows one event that is detected throughout all dataset-method combinations and corresponds with the highest number of hazard records over the analysed period (Figure 5). The Vaia storm hit the study region from 27[th] to 31[st] October 2018 and, based on in situ observations, led to exceptional cumulated precipitation over the five days of the event, exceeding 650 mm at Plöckenpass/Passo di Monte Croce Carnico in the Carnic Alps (Hübl and Beck, 2019). This local maximum is clearly depicted in all four datasets, but with spatial differences. INCA and SPARTACUS-TST agree on the spatial structure of the precipitation

field with lower amounts in the east of Carinthia and the highest precipitation totals in the western part of Carinthia and north of South Tyrol. However, INCA reports in general more detailed spatial gradients and higher precipitation amounts than SPARTACUS-TST, especially in South Tyrol. In contrast, the precipitation field presented by CERRA-Land differs significantly. Although it accurately captures the location of the precipitation minima and maxima over the domain, the spatial patterns appear noisier than the observational datasets. ERA5-Land, on the other hand, displays a much smoother precipitation

field, with values below 300 mm throughout the domain, due to its coarser spatial resolution, and reports a different spatial precipitation pattern in South Tyrol with respect to the other datasets. The coarser topography in ERA5-Land grid leads to a smoother and more spread precipitation fields over the domain, with intensity peaks less evident than in the other datasets. Regarding the registered hazards, it is noteworthy that most events are documented in the areas with the highest precipitation values, particularly in the south of Carinthia, even though some records are in areas where lower precipitation amounts are

reported by the gridded datasets.



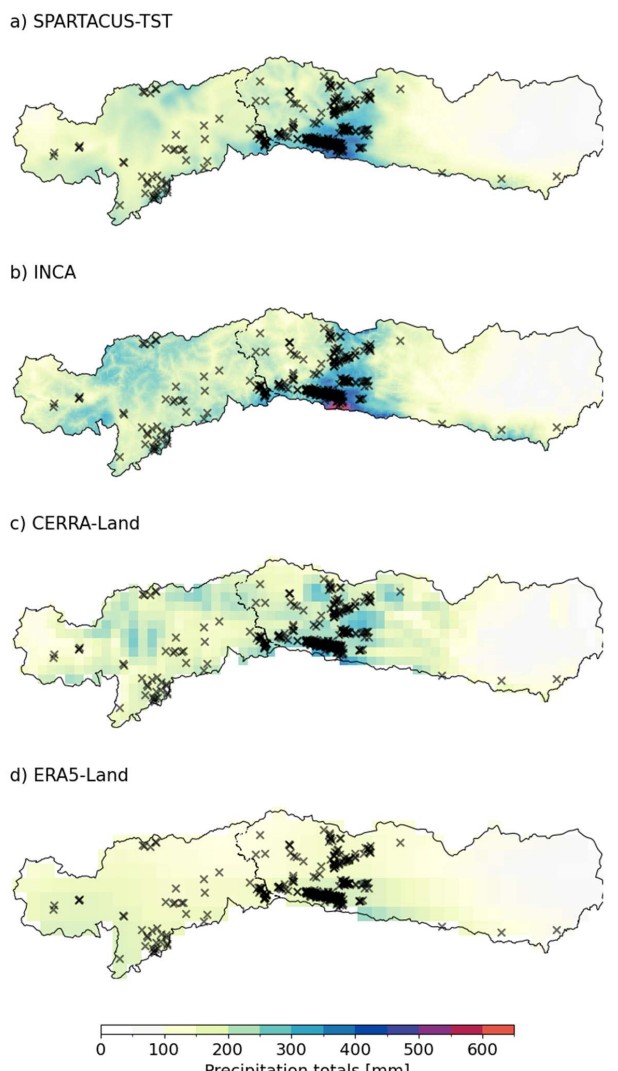

**Figure 8:** Cumulative precipitation over the Vaia event, spanning from 27th to 31st October 2018, based on (a) SPARTACUS-TST, (b) INCA, (c) CERRA-Land, and (d) ERA5-Land. All hazard records in the study region that occurred in that 5-day interval are indicated by crosses.

The distribution of recorded hazards with respect to the precipitation intensity, represented by different percentile classes, is used to further investigate the link between the regional spatial patterns of detected precipitation events by each dataset-method combination and the actual location of hazard records (Table 3). The total number of hazard records collected within the 5-day windows of each extreme precipitation event is also counted and considered for the comparison. Regarding the distribution of hazard records over precipitation classes, no significant differences are observed: for all datasets and methods, the number of

detected hazards increases with the precipitation class. Similar results are obtained if the Vaia storm (from 27th to 31st October 2018), for which the largest number of hazards (almost 400) was recorded, is excluded from the event samples. Almost all datasets capture the highest portion of intersected hazards in the highest precipitation class, especially ERA5-Land for all methods and CERRA-Land for the areal mean and the anomaly methods. This pattern can largely be attributed to the coarser resolution of these two products, which means that for large-scale events more hazards are likely to fall within the same,

relatively large, grid cells that belong to the high percentile class. This resolution effect underscores the importance of



considering grid-cell size and scale when interpreting the ability to capture hazard occurrences using precipitation data. The 330 extreme precipitation events defined by the local p99 method intersect the highest overall number of hazards across all datasets. In particular, precipitation events derived from INCA account for the greatest number of total intersected hazards (2,504). Precipitation extremes from reanalyses are generally those associated to the lowest amount of hazard records, even though the amounts of intersected hazard records by CERRA-Land fields are only slightly lower than the ones intersected by SPARTACUS-TST and even higher when the local p99 is considered. It is interesting to note that the portion of hazards in the greatest precipitation class is the highest for CERRA-Land across all methods.

| | | Percentile range | | | | | | |
|---|---|---|---|---|---|---|---|---|
| | | [0-0.1) | [0.1-0.3) | [0.3-0.5) | [0.5-0.7) | [0.7-0.9) | [0.9-1] | Total |
| Areal mean | SPARTACUS-TST | 4 % | 9 % | 13 % | 16 % | 31 % | 27 % | 1,732 |
| | INCA | 3 % | 8 % | 15 % | 17 % | 24 % | 33 % | **2,023** |
| | CERRA-Land | 1 % | 5 % | 11 % | 15 % | 26 % | 42 % | 1,648 |
| | ERA5-Land | 2 % | 7 % | 12 % | 24 % | 19 % | 35 % | 1,622 |
| Local p99 | SPARTACUS-TST | 4 % | 8 % | 12 % | 15 % | 31 % | 31 % | 1,800 |
| | INCA | 3 % | 6 % | 12 % | 28 % | 20 % | 32 % | **2,504** |
| | CERRA-Land | 3 % | 4 % | 15 % | 14 % | 28 % | 37 % | 1,964 |
| | ERA5-Land | 2 % | 7 % | 12 % | 22 % | 23 % | 33 % | 1,693 |
| Anomaly | SPARTACUS-TST | 3 % | 8 % | 10 % | 15 % | 33 % | 31 % | 1,748 |
| | INCA | 2 % | 8 % | 15 % | 15 % | 26 % | 33 % | **1,830** |
| | CERRA-Land | 3 % | 5 % | 10 % | 17 % | 26 % | 40 % | 1,690 |
| | ERA5-Land | 2 % | 7 % | 11 % | 23 % | 20 % | 36 % | 1,598 |

**Table 3:** Distribution over different precipitation classes of hazards recorded in a 5-day window of the top 330 (5 %) events identified for each dataset-method combination. Precipitation classes are defined as percentile ranges of the gridded precipitation values over the study area. Values are reported as percentage of the total intersected hazard records (last column). The dataset-method combination intersecting the highest total number of hazards is reported in bold.

By directly comparing the intensity of local precipitation at the locations of hazard occurrences, the different features of the gridded precipitation datasets emerge more clearly. Figure 9 illustrates the distribution of mean daily precipitation totals in the vicinity of hazards based on the events detected by the local p99 approach. The averages of intensities associated to hazard locations recorded on the same date are here represented to prevent the resulting distribution from being influenced by the different spatial granularity of the climate datasets, with multiple hazards falling on the same grid cell in the coarser products. For the reanalysis datasets, ERA5-Land shows mean precipitation intensities near hazards reaching up to 76 mm, with a median of approximately 20 mm and a 75th percentile around 29 mm. In comparison, CERRA-Land reports values as high as 100 mm, with the median and 75th percentile at roughly 22 mm and 34 mm, respectively. The high-resolution observational products show higher precipitation intensities. Precipitation distributions for SPARTACUS-TST and INCA are comparable in terms of median (26 mm and 24 mm, respectively) and maximum values (157 mm and 151 mm, respectively), with slightly higher intensities for SPARTACUS-TST. However, INCA reports a higher value of the 75th percentile and reflects the slightly different shape of the two distributions, with more values in the upper tail. This is also confirmed if the intensities in the proximity of hazards for SPARTACUS-TST and INCA are compared without averaging over the hazard records on the same date (Figure S5). INCA captured the highest local intensities, up to 246 mm, while SPARTACUS-TST reaches 169 mm only. The detailed statistics of precipitation values in the vicinity of hazards for all approaches are reported in Table S2. Similar results are observed for the precipitation events derived from the other two methods used in the study (Figure S6).

Similar conclusions can be derived also from the seasonal comparison. The summer half year distribution closely resembles the annual pattern for all methods, suggesting that precipitation extremes from April to September leading to geohydrological



processes substantially influence yearly precipitation statistics. In contrast, precipitation intensities spanned by the winter half year distributions are more moderate across all datasets, but still with the highest intensities captured by the observational datasets (Figure S7).

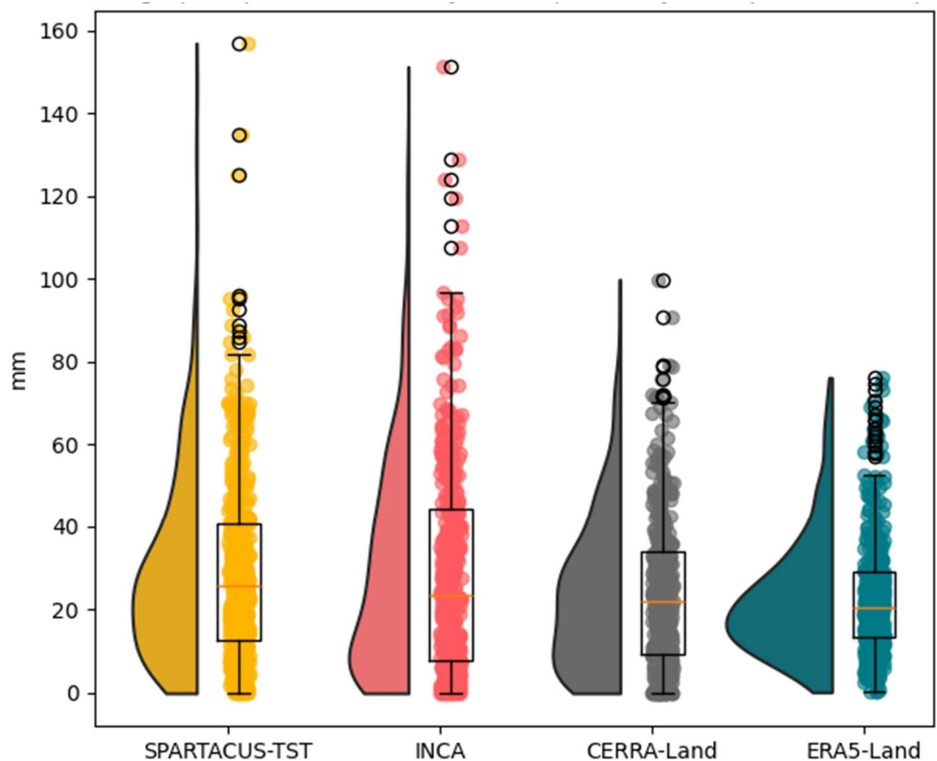

**Figure 9:** Distribution of daily precipitation intensities in the spatial proximity of recorded hazards for all datasets based on the most intense
5 % precipitation events (330 events) detected by the local p99 method. The precipitation intensities are extracted from the closest grid point to each hazard location and averaged over all hazard records on the same date.

**4 Discussion**

By comparing three different statistical methods for extreme detection and four target precipitation datasets, this study aims to provide insights into the suitability and limits of different climate products to identify and characterize hazardous precipitation
events in the period 2003-2020 over a transboundary and orographically complex area in the eastern European Alps. Moreover, recognizing the similarities and complementarities of different statistical definitions for daily extremes can enhance the understanding of how methodological choices influence extreme event detection. In particular, the main aim of the presented study was to evaluate the most suitable combinations of statistical definitions and datasets for the identification of the most extreme precipitation dates over the recent decades, the characterization of their spatiotemporal features and the intersection
with geohydrological hazard records. It is important to note that the findings rely solely on daily precipitation totals used as proxy for triggering effects of geohydrological hazards, described by recorded event days from hazard catalogues. In-depth analyses of sub-daily precipitation extremes and pre-moistening conditions are not explicitly included and will be addressed in future studies.

One of the most noteworthy considerations is that all combinations of datasets and statistical definitions yield similar temporal
patterns and seasonality of precipitation events. In particular, the most intense events over 2003-2020 are consistently detected across all approaches and datasets. This consistency highlights that, despite differences in the type and spatial resolution of



climate datasets and the specific extreme metric considered for event detection, the most intense precipitation extremes, especially when a substantial portion of the study domain is affected, are pointed out. However, differences emerge in precipitation features described by the datasets considered. The 1-km observation-based products show more highly resolved
spatial patterns of precipitation intensities over the domain with respect to the coarser reanalyses. The smoothed precipitation gradients reported by ERA5-Land make it the least suitable choice to derive a precise representation of precipitation intensities and local peaks. Interestingly, it reports the weakest annual cycle of precipitation over the region, while CERRA-Land, although sharing the same ERA5 boundary conditions, exhibits the strongest cycle and emphasizes summer precipitation totals. This underscores the effective contribution of the higher-resolution modelling scheme and the assimilation system of in situ
observations underlying CERRA-Land. The increasing trends of precipitation statistics over 2003-2020 reflect the temporal consistency among the datasets, even though products derived from operational forecasting, such as INCA, should be used carefully for this purpose as they are not designed for climatological analyses and can reflect potential changes in the production system over time.

The comparable rise in the number of geohydrological hazards recorded over 2003-2020 by the Austrian and Italian authorities,
suggests that the increase of precipitation extremes contributed to increase the probability of hazardous phenomena (hence leading to more recorded hazards). However, the rigorous identification of a causal link between the intensification in precipitation statistics and the increase of hazard records is limited by the short length of data as well as by the potential influence of non-climatic factors which can contribute to the hazard record variability, such as changes in the procedure of hazard record collection over time. As hazard occurring near infrastructure and causing damage are generally more likely to
be recorded, also the increased exposure from expanding infrastructure can play a role. Future studies should consider exposure and vulnerability as essential variables to get a more comprehensive explanation of the temporal trends of hazard records (Schlögl et al., 2021).

The study highlights how a preliminary evaluation of precipitation data and hazard records is needed to understand the most suitable criteria to apply for their analysis and intersection. Potential mismatches in the temporal alignment of daily
precipitation records among different precipitation datasets and with hazard databases made it necessary to use a multi-day window to distinguish precipitation events and analyse the temporal intersections with hazard occurrences. The choice of a 5-day window was supported by autocorrelation and cross-correlation analyses. The high frequency of hazard occurrences over the study period, i.e., the relatively high frequency of days with at least one hazard event recorded, also required filtering out the hazardous dates to retain less likely events characterized by a higher number of recorded hazards to compare with extreme
precipitation dates. This stems from the consideration that isolated 'background' hazards are less likely to be triggered by extreme precipitation and rather be connected to other processes and environmental conditions on different time scales. Although arbitrary, the threshold used for filtering the hazardous dates (at least two recorded hazards over the domain) is justified by the need of not reducing too much the sample size while retaining the most relevant subset of hazard records. The same criterium was also applied for the selection of the most extreme precipitation events, by limiting the test sample to 330
dates, which correspond to about the top 5 % of the entire precipitation series. Future studies should include a more systematic evaluation of how the choice of the threshold influences the explained portion of all hazardous dates (Spiekermann et al., 2023). This is expected to provide valuable insights for refining the evaluation and the definition of precipitation extremes.

The joint analysis of meteorological events and recorded hazards performed in this study shows a moderate relationship between extreme precipitation and hazard occurrence, which is independent from the considered combination of datasets and
detection methods. While it is found that extreme precipitation days increase the likelihood of recording hazards by 7 to 9-fold in all cases, the relatively low correlation and the hit rate, i.e., the portion of hazardous dates intersected by the top 5 % precipitation extremes, below 50 %, clearly indicates that other factors besides the daily precipitation intensity contribute to



explain the occurrence of geohydrological phenomena. Nevertheless, the comparison of hit rates obtained by each combination of datasets and methods reveals non negligible differences in the detection skill, even though it is not possible to uniquely conclude which choice performs best for identifying hazardous episodes. Regardless of the method used, coarser spatial resolution products, such as ERA5-Land reanalysis, have shown a limited ability to identify precipitation extremes likely associated with local hazard occurrences. In contrast, higher resolution products, especially those incorporating observations, offer a better accuracy in detecting extreme precipitation events with hazard potential, although such products are generally available only at a national level, which poses challenges for cross-border analyses. A direct combination of different national products might be hampered due to differences in the used interpolation methods or inconsistencies in underlying data coverage, hence requiring careful investigation. Additionally, the limitations of spatially interpolated station data must be considered since phenomena occurring at scales smaller than the mean inter-station distance, such as small-scale convective storms, may not be captured, leading to under-representation of localized extreme events (Hiebl and Frei, 2018). Based on the observed performances, high-resolution multi-source datasets such INCA integrating rain gauges, radar estimates and model simulations, can better support the detection of small-scale, high-intensity events. However, for products like INCA optimized for operational forecasting application, the analysis of temporal variability (Sect. 3.1.1) suggests that caution is required since they could be more prone to potential inhomogeneities due to changes in the forecasting system over time. Notably, CERRA-Land, showing hit rates comparable to those of observation-based datasets, proved to be a suitable alternative for deriving extreme precipitation events with hazard potential in the transboundary area. The 5.5-km grid and the integration of rain gauge data in the precipitation assimilation system contribute to the better performance in representing precipitation extremes and their timing with respect to the coarser ERA5-Land reanalysis, where in situ observations are not directly used.

These results confirm the findings from other inter-comparison studies focusing on the characterization of precipitation extremes and expand them by integrating the comparison with hazard records. For instance, Hu and Franzke (2020) showed for Germany that the best representation of the magnitude of 1-day precipitation extremes is achieved by the high-resolution national observation dataset. In contrast, both ERA5 and the regional COSMO-REA6 reanalyses, lacking a proper assimilation scheme of precipitation data, report the least accurate statistics, especially regarding the timing of extremes. Extending the assessment to other Alpine regions could be useful to confirm the skill of CERRA-Land and may have important implications for transboundary studies. The availability of high-resolution accurate reanalysis fields overcomes limitations due to missing cross-boundary observation products or heterogeneity in national/regional datasets.

Among methods, extremes detected based on a high percentile calculated over the whole study domain were found to be more linked to hazardous dates than those identified through approaches more influenced by the extent of precipitation events. This is likely because the percentile-based method accounts for very intense but more localized precipitation events which are ranked lower by the other areal-based methods but still recorded local hazards. This suggests that local events can trigger disproportionally more hazard phenomena than precipitation events with a larger extent but lower intensities. Areal-based definitions might be preferable when the aim is to detect the main events driven by large-scale precipitation systems, especially for wider regions characterized by more heterogeneous climatic regimes and topographic patterns, e.g., the whole Alpine region. Sensitivity tests evaluating the performance of extreme definitions based on the targeted spatial scale and types of precipitation phenomena could provide a more detailed guidance for better tailoring hazard and impact-oriented studies.

The portion of hazardous dates over 2003-2020 included in the top-ranking precipitation extremes below 50 % may suggest that 1-day precipitation intensity is not enough to explain hazard occurrence. For instance, some hazard records might be better correlated to hourly peak rainfall intensities which are expected to be under-represented by only considering the daily sums. Moreover, other factors, including preparatory conditions, concur to influence some hazardous geohydrological processes, even with low-intensity precipitation events. Such factors include for instance temperature anomalies, free-thaw cycles, soil



moistening and snowmelt, which are not considered here (Banfi and De Michele, 2024). This might partly explain the reduced
hit rates for the winter period, where the intensity of precipitation phenomena over the study domain is generally lower than
in summer.

The spatial analysis revealed that the comparable performance of datasets in the temporal alignment of hazardous dates with
precipitation extremes is not equally reflected in the analysis of spatial relationships between precipitation intensities and
hazard locations. In particular, the representation of precipitation intensities in the spatial proximity of hazard occurrences was
found to be primarily dataset-dependent rather than method-dependent. Although hazards are predominantly found in areas
with higher precipitation intensities, spatially refined datasets can resolve higher precipitation peaks in the proximity of
hazards. In contrast, both reanalyses show a similar under-representation of local 1-day precipitation amounts, regardless of
their different spatial resolution. This finding suggests that the choice of the product is key when the analysis requires a detailed
description of precipitation intensities and small-scale aspects of triggering phenomena to be linked with hazard occurrences.
In the example of the Vaia storm event (Figure 8), the smoother precipitation fields of ERA5-Land are clearly the least suitable
to analyse and model the spatial correspondences between precipitation intensities and hazard occurrences. However, also the
presence of local systematic biases due to the uneven spatial distribution of observations underlying the climate datasets should
be investigated to ensure the robustness of results. Another limitation in this study to be considered, next to precipitation
datasets and detection methods, is related to the hazard records. Only some types of weather-induced hazards were considered,
i.e., floods-related processes (e.g., flooding, surface runoff, fluvial sediment transport) and gravitational mass movements (e.g.,
debris flows, mud flows, translational/rotational slides). Since hazards are generally recorded when road and infrastructure
incur damage and necessitate intervention, the merged database of hazard observations might be negatively biased in remote
and unsettled areas, thus underrepresenting all actual hazardous mass movements and flooding processes occurred in the region
and potentially triggered by extreme precipitation episodes (Schlögel et al., 2020; Steger et al., 2024).

**5 Conclusions**

This study provides a first assessment of methods and meteorological datasets for the detection and characterization of extreme
precipitation events possibly triggering geohydrological hazards in a transboundary Alpine area covering South Tyrol in Italy
and East Tyrol-Carinthia in Austria. Daily precipitation fields from two observation-based (SPARTACUS-TST and INCA)
and two reanalysis products (CERRA-Land and ERA5-Land) were considered along with a composite database of recorded
hazards (mostly floods and mass waste phenomena) collected from several national sources.

The most relevant precipitation extremes occurred over 2003-2020, a significant increase in daily precipitation intensities, and
the higher frequency in summer are consistently detected by all datasets and methods. However, there are clear advantages
and shortcomings of some dataset-method combinations over others, which necessitates careful selection based on the goal of
the analysis. The precipitation events defined by a percentile-based approach using local intensities (local p99) coincide with
more dates registering hazards, especially when the method is combined with the INCA dataset merging in situ observations
with radar measurements. The intersection with hazards slightly decreases for SPARTACUS-TST and CERRA-Land, which
remain still robust alternatives to the multi-source INCA data, while it is substantially lower for ERA5-Land based on all
methods, confirming the limitation of the 9-km reanalysis in describing precipitation extremes with hazard potential in the
study area.

While the spatial relation between precipitation intensities and the hazard occurrences is consistent across all dataset-method
combinations, the description of precipitation peaks in the spatial proximity to hazards is largely determined by the dataset
used. The highest intensities are captured by the multi-source INCA dataset, especially for events detected by the local p99
definition. CERRA-Land still benefits from the higher spatial resolution and precipitation data assimilation compared to



ERA5-Land but still lags behind the 1-km observation-based regional datasets, which remain the primary choices when applications rely on an accurate description of precipitation fields.

The study also showed that gridded observational datasets of nearby regions are generally not directly usable in a merged version and some preliminary steps are needed to increase their consistency (e.g., spatial and temporal alignment), even though some discrepancies cannot be fully removed (e.g., due to different interpolation methods and observations used). The development of harmonized and high-resolution precipitation datasets using dense in-situ observations and crossing multiple

regions remain thus key for transboundary analyses on climate-related hazards and risk management. In future studies, the evaluation can be further extended by including km-scale reanalyses, which explicitly resolve convective processes, to assess their potential added value for hazard and impact-oriented analyses, although these products are generally available either at a national scale, or for short periods (e.g., Frank et al., 2020; Giordani et al., 2023). Similarly, the integration of larger-scale gauge-adjusted radar precipitation products, such as EURADCLIM for Europe (Overeem et al., 2023), might provide

additional insight into their applicability for cross-border analyses of weather extremes and natural hazards.

While more systematic evaluations of existing precipitation data sources and event detection algorithms are needed, the current findings demonstrate that even relatively simple statistics combined with sufficiently accurate data products provides meaningful information about spatio-temporal patterns of precipitation extremes with hazard potential. The three extreme detection methods require only daily precipitation estimates as inputs and adopt an event ranking algorithm using extreme

value definitions based by simple statistical assumptions, which makes these approaches directly applicable to different types of meteorological products and to different spatial domains. Depending on the event type of interest, e.g., large-scale phenomena or small convective episodes, and the available climate datasets, the extreme definitions can be used individually or in combination to extract the spatio-temporal features of precipitation extremes and compiling comprehensive event portfolios supporting further risk analyses in cross-boundary regions. This study, and the released event database, can inform

methodological choices for the development of impact-based early warning systems and the implementation of risk-oriented applications in cross-boundary regions.

**Code Availability**

All codes are available upon request from the authors.

**Data Availability**

The top 5 % extreme precipitation events identified by each method-dataset combination and the full daily precipitation statistics over 2003-2020 from all climate datasets are available from Crespi et al. (2025): https://doi.org/10.5281/zenodo.15756269. Hazard records are available upon request.

**Author Contribution**

AC and EK collected the data, performed the analyses and discussed the results. LS, KH and MP contributed to methodology

and result discussion. MP provided support for the analyses and supervision. AC and EK wrote the papers with contributions from all co-authors.

**Competing interests**

The authors declare that they have no conflict of interest.



**Acknowledgements**

The regional agencies (Austrian Service for Torrent and Avalanche Control, the Autonomous Province of Bolzano and the Geological Survey of Austria) and the Italian National Institute for Environmental Protection and Research managing the hazard databases used in this study are acknowledged. The Copernicus Climate Change Service providing access to the reanalysis data is acknowledged. The TransAlp project ("Transboundary Storm Risk and Impact Assessment in Alpine regions") financed by the European Commission under the program UCPM-2020-PP-AG (Prevention and Preparedness

Projects for Civil Protection and Marine Pollution) is acknowledged for initiating the discussion leading to this work.

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
