# Peer review of "Detection and characterization of precipitation extremes and geohydrological hazards over a transboundary Alpine area based on different methods and climate datasets"

_EGUsphere, 2025_

## Referee Comment (RC2)

Crespi and Enigl et al test the suitability of four different meteorological datasets and three different extreme event metrics for the detection of observed geohydrodological impacts, namely floods and mass movements. They study the temporal consistency of extreme precipitation events extracted from the four datasets with the impact events, and further try to show spatial consistency as well. The analysis is done for a transboundary domain in the European Alps.

I think that the question on which datasets can be used for these types of events as well as which extreme precipitation metric can detect these events is relevant.

I think one limitation in the analysis is that floods and mass movement events are pooled together in the analysis. While I understand that both events can be triggered by extreme precipitation, I do think that it would be worth while also analyzing the dependence of your results on both of these events independently. The reason for this is, that the low "hit rate" of maximum 50% could be related to that only one type of event is well represented, and the other is not. I think there would be merit in giving recommendations on whether the dataset and methods work for floods or mass movements, or both or both equally good/bad.

The methods section generally needs more clarification of the individual steps and should also include some justification of the method choices. Some of the contents should be moved to the results section.

Generally, it is not always easy to follow, and I would recommend working a bit more on the flow of the text and potentially simplifying the sentences. You partly use very long sentences with a lot of information. Further, I think in the results part some sections are quite long. I would suggest splitting these into multiple parts. This will improve the flow of the text and will make the results more accessible to the readers.

Lastly, I think the conclusions can be condensed quite a bit and suggest to focus on the main results of your analyze.

**Abstract:**

While the abstract is generally well written, the sentences are very long containing a lot of information. Try to rework the sentences in a way that you maintain good flow but reduce the sentence length to be more accessible to non-experts. I would maybe mention which four meteorological datasets you used, but at least mention the different types, e.g. gridded observations, reanalysis, radar based.

**Introduction:**

P1 L 35: "hydrometeorological events" -> can you mention which ones?

P2 L 50-68: Three remarks: 1) I think the first part on the definition of extremes is quite generic. You have two very clear impacts that you are trying to link to precipitation extremes. Therefore, I would suggest that you try to be more specific and focus on what event definitions have been used in the context of floods and mass movements. Make this first part its own paragraph. 2) I would merge second half of the paragraph (starting in L 58 "Moreover, ...") with the next paragraph. 3) "Seneviratne et al 2021" is not included in the reference list.

P2 L 77: Wood et al (2024) -> replace with published version

P 2 L81f: Be more specific here. Define clear research questions and briefly describe how you will answer these questions (e.g. To answer these questions we compare four datasets of different complexity and three different extreme event metrics ....)

**Material and Methods:**

**Study area:** Why did you limit your analysis to this study region? From the data description it sounds like you could have extended your analysis also to other Austrian regions. Maybe just include one short sentence on this.

P 4 L 130-32: "The temporal dimension ..." -> How was this achieved? Was the TST dataset available in higher temporal resolution and then daily totals have been re-calculated?

**INCA**: Mention in the description that only very few stations were included in South Tyrol and that there are no radars on the Italian side.

P5 L163: "primary advantage" of CERRA-Land -> I would mention here that CERRA-Land is also one of the only reanalyses that assimilate precipitation gauge data. Except for ERA5 over the US.

P5 L 169: "lapse rate correction" -> I think that the lapse-rates correction was only applied to temperature and not to precipitation. Precipitation is only linearly interpolated to the higher resolution.

P6 L181f: This and the following paragraphs are part of the results section. You could add a new first result section titled "Comparison of general extreme precipitation statistics". Further, since your study is mainly motivated by extreme precipitation, wouldn't it be more meaningful to show some metric that represents extreme precipitation in Figures 2 and 3? In Figure 2, for example monthly 99th percentile or monthly max precipitation. You can place the current figure on mean seasonality in the supplement and briefly mention it in one sentence. The same would then apply to Figure 3.

P6 L 190: "SPARTACUS-TST and INCA are comparable" -> I would maybe mention that INCA is considerably and systematically lower than SPARTACTUS-TST over South Tyrol, which is likely due to the limited number of stations used and no radars. Also west of Lienz, the precipitation is lower in INCA. I would suspect that the radars are blocked in the north and east by topography.

P 7 L202: "processes are closely linked to extreme precipitation events" -> For floods this is quite obvious, but could you include some studies that show the connection between extreme precipitation and mass movements.

**Figure 4 & 5:** I would suggest merging both figures and slightly change the contents. Have Fig 5 as new panel a, and then figures 4 below as panels b and c. Would it be possible to have the Yearly and monthly distributions as stacked bar plots consisting of the two types of hazards (floods and mass movements).? Also adapt the new figure caption giving a descriptive title first before describing the contents of the panels.

**Methodology:**

This section is currently difficult to follow, and the methods described need a bit more clarification and a justification of the methods choices. It is very important that everyone understands the event definition and selection.

P10 L 284: "the available precipitation data..." -> Why this assumption? With most of your datasets you could test this assumption.

P11 L 296: "The ranked values ..." -> I would remove this sentence since you are not doing this.

P11 L 304: "is expected to capture ..." -> Doesn't this bare the risk of only sampling events that are located in the "high precipitation" areas?

P11 L314: "all wet-day values" -> Have wet days also been excluded from the other two methods? Or is it uncommon that the entire region has zero precipitation?

P11 L317: "The product of the two..." -> Please be more explicit how you combined these two metrics.

P11 L322-23: "To ensure that ... is retained." -> Does it make a difference whether you do the event filtering (i.e. clustering events within a 5-day window) before or after the ranking? When you remove the clustered dates from the ranked list, do you adjust the rank of the remaining list? Do you remove these dates entirely from all analysis?

P11 L 324: "top 5% of sorted dates" -> Why the top 5%? And this 5% applies to the ranked list where dates belonging to the same event "within 5-days" have been removed?

P12 L 344-45: "A lower threshold ..." -> Did you also consider days with single hazards but clustered in time and space. Meaning that hazards from the same storm (say 3-days long) may trigger single hazards on each of these days in close spatial proximity. At the moment these hazard events would not be accounted for even though you might account for the 3-day precipitation event.

P12 L 356f: "Since ..." -> Create a new subsection. This will break up the methods description and it is easier to follow. This section would only cover the "spatial coherence" analysis. The previous one "hit and miss".

P12 L 365: "To achieve this ..." -> On which basis are these percentile classes calculated? based on "all days" in the period 2003-2020, based on "wet days" in the period 2003-2020, or based on "extracted precipitation events" only?

P12 L366: "Each hazard record ..." -> Is the "four nearest grid cells" applied irrespective of the spatial resolution? Isn't this likely penalizing the higher resolution? While the 1km grids suggest a higher accuracy, these datasets still have an effective resolution of 10-15km.

**Results**

Have you also tested your results independently for the two hazard types (floods vs. mass movements)? I am wondering whether we can say something about whether the hit rate for floods is better than for mass movements. Both can be connected to intense precipitation, but I think it would be valuable if you could say floods are detected in x% of the cases and mass movements in y%. I am simply wondering whether the low hit rate (i.e. 50%) is due to the inability to match mass movement events which are very localized events compared to some of the flood events. For these events likely non of the datasets might be suitable.

I think generally I would deemphasize the trend analysis. Detecting trends from the short time period and the shortcomings of the datasets inhibits any trustworthy trends.

**Section 3.2.2:** I think the link between temporal hits (based on the areal statistics) and the link to the spatial hits can be strengthened. It would be interesting to know how many of the correctly detected hazard days (up to 50%) also show a correct spatial detection. Meaning that within your temporal search window and your spatial window you have an actual precipitation

value which qualifies as an extreme value. You do analyze the connection already, but you could maybe make this a bit more implicit defining a spatial "hit rate".

P13 L 387: "INCA is characterized by more pronounced increases in all statistics" -> This might be due to the inclusion of new radars in the recent years....

P14 L415: "72% to 88%" -> Are these numbers irrespective of the ranks, meaning that they agree on common unique days but can show inversed ranks? Did you also check the agreement in the rank locations of the events? Did you also quantify the agreement across the three event metrics?

P14 L418-21: "For all methods ...." -> Where can we see this? How large is the overlap between the different datasets?

P16 L463: "correlation precipitation statistics and hazards ...." -> Are these correlations with all hazard days or with only hazard days with at least two reportings? I thought in your methods section you explained that you remove all days with only a single hazard reported.

Figure 8: Could you maybe add the statistics of the three metrics for this event and their ranks for each of the datasets.

P18 L516-18: "Almost all datasets ..." -> Mention SPARTACUS-TST as an exception here. This dataset shows almost equal proportion of events in the second highest class (0.7-0.9), which represents more a moderate event intensity. My hypothesis would be that this is connected to the rather strict spatial rule of 4 closest grid cells and the inherent precipitation smoothing between stations. So, I think if you would extent the search radius to the scale of ERA5-Land (approx. 9 km) then the match to the highest precipitation class might be larger.

P19 L543: "more values in the upper tail" -> but also the lower tail. INCA shows a stronger left skewed distribution, which means that in INCA we have quite a few events with very low intensities.

Figure 9: Can you change the line color of the median to black. The contrast is poor for the salmon color (INCA). You can also remove the unfilled outliers from the boxplots (non filled black circles), since these data points are shown in color anyway. Wouldn't it be better to simple show the precipitation intensities for all hazard events? As I understand you plot the distribution based on the p99 event selection with a hit in hazard, meaning that each distribution is based on a varying number of events. Ranging between 1693 to 2504 hazard events. Or am I interpreting this wrong?

**Discussion**

P21 L580-83: "The increasing trends of precipitation ..." -> I would rephrase this sentence and focus on the temporal consistency rather than mentioning trends. The period is too short to really say something about trends and in two out of the four datasets you have limitations in the consistency of the dataset.

P21 L584: "rise in the number of geohydrological hazards..." -> I think you need to mention here again the limitation of hazard record, which is likely strongly affected by reporting bias. I think you mentioned this is the methods section.

P21 L603-04: "Although arbitrary, ..." -> Have you considered to also filter by hazards in proximity, but one day apart?

P22 L629-31: "The 5.5km grid and ..." -> You can mention here Wood et al 2025 again to show that this is a consistent finding.

P22 L649-51: "The proportion of hazardous dates ..." -> It might also suggest that the extreme event metrics are not capturing the essence of these events.

"For instance, ...." -> From at least one of your datasets (INCA) and potentially TST (if it is an hourly datasets) you could test this hypothesis. You could aggregate the hourly data by taking the daily maximum instead of the daily sums. Then do your analysis accordingly and check whether the detection rate is higher or lower.

P23 L663-64: "This finding suggests..." -> However, as I mentioned before we can see for INCA many instances where precipitation in the vicinity of the hazard is very low (left skewed distribution).

**Conclusions:**

Shorten the conclusion to the most important take-aways that answer your research questions. You mention several details which you didn't really analyze and which are part of the methods and not a result of your comparison.

P23 L680: "mass waste" -> do you mean "mass movements"?

P23 L681: "a significant increase in daily precipitation intensities" -> not really relevant and not a key finding of your study. I would remove this.

P24 L696-98: "The study also showed ..." -> Not relevant, can be removed.

P24 L700-05: "In future studies, ...." -> Move this to the discussion section

P24 L708: "meaningful information about ..." -> This statement contradicts a bit your finding of only 50% of hazards being detected by the extracted precipitation events.

---

## Author Comment (AC1)

**Detection and characterization of precipitation extremes and geohydrological hazards over a transboundary Alpine area based on different methods and climate datasets**

GENERAL COMMENTS

Crespi and Enigl et al test the suitability of four different meteorological datasets and three different extreme event metrics for the detection of observed geohydrodological impacts, namely floods and mass movements. They study the temporal consistency of extreme precipitation events extracted from the four datasets with the impact events, and further try to show spatial consistency as well. The analysis is done for a transboundary domain in the European Alps.

I think that the question on which datasets can be used for these types of events as well as which extreme precipitation metric can detect these events is relevant.

I think one limitation in the analysis is that floods and mass movement events are pooled together in the analysis. While I understand that both events can be triggered by extreme precipitation, I do think that it would be worth while also analyzing the dependence of your results on both of these events independently. The reason for this is, that the low "hit rate" of maximum 50% could be related to that only one type of event is well represented, and the other is not. I think there would be merit in giving recommendations on whether the dataset and methods work for floods or mass movements, or both or both equally good/bad.

The methods section generally needs more clarification of the individual steps and should also include some justification of the method choices. Some of the contents should be moved to the results section.

Generally, it is not always easy to follow, and I would recommend working a bit more on the flow of the text and potentially simplifying the sentences. You partly use very long sentences with a lot of information. Further, I think in the results part some sections are quite long. I would suggest splitting these into multiple parts. This will improve the flow of the text and will make the results more accessible to the readers.

Lastly, I think the conclusions can be condensed quite a bit and suggest to focus on the main results of your analyze.

*We thank the reviewer for taking the time to read the manuscript and providing useful comments and suggestions. Based on them and on the feedback provided by the other reviewer, the main changes in the revised manuscript are:*

- *We restructured Data and Methodology sections by moving part of the contents to Results. In particular, the comparison among the datasets and the overview of collected hazard records are now in the new subsection 3.1 ("Precipitation statistics from meteorological datasets and hazard record overview").*
- *We considered also hazard types (i.e., flood/mass movement) to integrate the hit rate analyses. In particular, we plotted the monthly and annual distributions per hazard type (now Figure 5) and we discussed the hit rates for each category, separately.*
- *We slightly revised the methods for hit rate calculation, precipitation class and intensity assignment as described below and in the revised manuscript.*

- *We updated the hazard dataset by integrating the new version of WLV and GERIOS, which slightly increased the number of hazard records in our set. We updated all numbers and results based on the updated version.*
- *We revised Discussion and Conclusions by shortening them and making key messages more prominent and better related to the research questions of the study.*

*Specific comments are addressed below.*

ABSTRACT

While the abstract is generally well written, the sentences are very long containing a lot of information. Try to rework the sentences in a way that you maintain good flow but reduce the sentence length to be more accessible to non-experts. I would maybe mention which four meteorological datasets you used, but at least mention the different types, e.g. gridded observations, reanalysis, radar based.

*We revised the abstract by splitting and restating some long sentences. Moreover, we explicitly mentioned the datasets used and their types (i.e., reanalysis, observation-based, radar-aided).*

INTRODUCTION

P1 L 35: "hydrometeorological events" -> can you mention which ones?

*We specified the main types of hydrometeorological events mentioned in the cited report of Munich RE for 2023 in Europe.*

P2 L 50-68: Three remarks: 1) I think the first part on the definition of extremes is quite generic. You have two very clear impacts that you are trying to link to precipitation extremes. Therefore, I would suggest that you try to be more specific and focus on what event definitions have been used in the context of floods and mass movements. Make this first part its own paragraph. 2) I would merge second half of the paragraph (starting in L 58 "Moreover, ...") with the next paragraph. 3) "Seneviratne et al 2021" is not included in the reference list.

*We revised the first part of this paragraph by referring to studies analysing extremes and, in general, precipitation triggering potential in the context of floods and mass movements (e.g., Barton et al., 2022; Meyer et al., 2022, Breugem et al., 2020). The second half of the paragraph was integrated into the subsequent one and additional references covering hazard-related studies are integrated (e.g., Peruccacci et al., 2017; Steger et al., 2023; Vaz et al., 2018; Araújo et al., 2022; Banfi and De Michele 2024). "Seneviratne et al. (2021)" is not cited anymore in the revised version of the text.*

P2 L 77: Wood et al (2024) -> replace with published version

*Done.*

P 2 L81f: Be more specific here. Define clear research questions and briefly describe how you will answer these questions (e.g. To answer these questions we compare four datasets of different complexity and three different extreme event metrics ....)

*We revised the last paragraph of the Introduction by elaborating specific research questions and mentioning how these questions are addressed by the study:*

*"In this framework, the study aims to i) evaluate how metrics for precipitation intensity, not a-priori tailored to a specific hazardous process, enable to capture extreme events with triggering potential for geohydrological hazards over complex topography; ii) assess the suitability of precipitation datasets of different types and spatial resolution to describe extremes; iii) investigate the optimal combinations of metrics and datasets for characterizing extreme precipitation events and their spatio-temporal relation with hazard records. To answer these questions, three metrics measuring different aspects of rainfall extremes are calculated from 1-day precipitation fields of four meteorological datasets over a transboundary Alpine area between Italy and Austria and used to identify precipitation events over 2003-2020. Subsequently, they are compared with a harmonized archive of geohydrological hazard records to quantify the spatio-temporal match between identified events and observed records.".*

MATERIAL AND METHODS

**Study area:** Why did you limit your analysis to this study region? From the data description it sounds like you could have extended your analysis also to other Austrian regions. Maybe just include one short sentence on this.

*The meteorological and hazard datasets we used would allow us to cover the whole of Austria. However, the analyses were focused on the transboundary subregion of South Tyrol and East Tyrol/Carinthia since they present a similar mountainous terrain which makes them similarly prone to precipitation-induced geohydrological hazards of interest. Due to the wide extent and orographic heterogeneity of Austria (including both mountains, hills and wide plain areas), the triggering role of precipitation for hazardous processes is expected to highly vary across the domain together with the types of induced hazardous processes and their spatial and temporal patterns, which may reduce the interpretability of results. For the purposes of the study, the most relevant aspect of the selected domain is its transboundary nature as it represents a critical aspect for risk assessment and management in Alpine regions. We added a sentence to motivate the choice of the study area in Section 2.1, accordingly.*

P 4 L 130-32: "The temporal dimension ..." -> How was this achieved? Was the TST dataset available in higher temporal resolution and then daily totals have been re-calculated?

*The TST dataset has a daily resolution. The daily precipitation totals in TST are defined as the cumulative precipitation from 8:00 UTC of the previous day to 8:00 UTC of the current day, as this is the definition of daily precipitation total adopted by the local provider of station records used for developing TST. SPARTACUS is also daily, but the daily totals are defined as the cumulative precipitation from 6:00 UTC of the current day to 6:00 UTC of the following day.*

*This generates the misalignment of the two datasets when merged, so that a 1-day shift is necessary to realign the temporal dimension. We restated the text in the manuscript to clarify better this aspect.*

**INCA**: Mention in the description that only very few stations were included in South Tyrol and that there are no radars on the Italian side.
*We integrated the description of INCA with more details about the underlying data used.*

P5 L163: "primary advantage" of CERRA-Land -> I would mention here that CERRA-Land is also one of the only reanalyses that assimilate precipitation gauge data. Except for ERA5 over the US.
*We added this point.*

P5 L 169: "lapse rate correction" -> I think that the lapse-rates correction was only applied to temperature and not to precipitation. Precipitation is only linearly interpolated to the higher resolution.
*We originally included this sentence to describe the overall ERA5-Land product. However, the reviewer is right that it is not relevant for precipitation and it has been removed.*

P6 L181f: This and the following paragraphs are part of the results section. You could add a new first result section titled "Comparison of general extreme precipitation statistics". Further, since your study is mainly motivated by extreme precipitation, wouldn't it be more meaningful to show some metric that represents extreme precipitation in Figures 2 and 3? In Figure 2, for example monthly 99th percentile or monthly max precipitation. You can place the current figure on mean seasonality in the supplement and briefly mention it in one sentence. The same would then apply to Figure 3.
*The entire paragraph was moved to the Results section and merged with the analysis of the temporal variability in precipitation statistics (now Section 3.1"). The trend analysis was downsized as we recognized that the 18-year series are not long enough for a robust estimation of trend, while the trend assessment performed over the longer period was kept in the Supplementary Material as it supports the signal emerging over 2003-2020. We modified Figure 2 and Figure 3 (see below) by replacing the monthly precipitation totals with the 99$^{th}$ percentile (calculated over the temporal and spatial dimensions) and updated the text accordingly. The previous version of Figure 2 was kept in the Supplementary Material (now Figure S2) and used to integrate the comparison of seasonality of 99$^{th}$ percentile. The assessment of hazard records originally in the Data section was also integrated into 3.1.*

[Figure]

**Figure 2:** Monthly 99[th] percentile of daily precipitation calculated over all days in 2003-2020 and all grid points in the study area for the four gridded datasets considered.

[Figure]

**Figure 3:** a) Winter half year (October to March) and b) summer half year (April to September) 99[th] percentile of daily precipitation totals over 2003-2020 in the study area based on SPARTACUS-TST, INCA, CERRA-Land and ERA5-Land. Each dataset is shown in its native spatial resolution.

P6 L 190: "SPARTACUS-TST and INCA are comparable" -> I would maybe mention that INCA is considerably and systematically lower than SPARTACTUS-TST over South Tyrol, which is likely due to the limited number of stations used and no radars. Also west of Lienz, the precipitation is lower in INCA. I would suspect that the radars are blocked in the north and east by topography.

*It is correct that the data basis in South Tyrol within INCA is less robust. In 2015, the Bolzano radar was integrated into the system; however, this radar has still been affected by recurring technical issues (e.g., outages). Regarding the issue in East Tyrol, it is indeed correct that this region is affected by shielding effects and is not adequately covered by either the Austrian radar network or the South Tyrolean radar. As a consequence, precipitation amounts in this area might be underestimated. However, in the revised version of the manuscript, the dataset comparison shown in Figure 3 focuses on the 99$^{th}$ percentile of 1-day precipitation totals for which this effect is not evident anymore, so we removed this previous sentence from the paragraph describing Figure 3. In any case, we mentioned possible underestimations in INCA fields due to radar shielding effects in other parts of the manuscript.*

P 7 L202: "processes are closely linked to extreme precipitation events" -> For floods this is quite obvious, but could you include some studies that show the connection between extreme precipitation and mass movements.

*We added a reference and slightly modified the sentence: "We focus on gravitational mass movements and floods, since the occurrence of these hazardous processes can be largely influenced by precipitation intensity (e.g., Borga et al., 2014)."*

**Figure 4 & 5:** I would suggest merging both figures and slightly change the contents. Have Fig 5 as new panel *a*, and then figures 4 below as panels *b* and *c*. Would it be possible to have the Yearly and monthly distributions as stacked bar plots consisting of the two types of hazards (floods and mass movements).? Also adapt the new figure caption giving a descriptive title first before describing the contents of the panels.

*We merged the two figures as suggested, but keeping the order of panels as in the original manuscript since the annual and monthly distributions are mentioned first in the text. The new Figure 5 includes panels a) and b) with annual and monthly distributions of hazards as stacked bar plots by distinguishing for hazard type and panel c) in the bottom reporting the time series of hazards previously in Figure 5. The caption of the new Figure 5 was revised accordingly and following reviewer's suggestions.*

[Figure]

**Figure 5:** Overview of documented geohydrological events in the study area (South Tyrol, East Tyrol and Carinthia) over 2003-2020 sourced from the IFFI and ED30 databases for South Tyrol and the WLV and GEORIOS databases for Austria: a) annual and b) monthly distributions of observed events distinguishing between flood and mass movement types and c) 2003-2020 daily series of the total number of events recorded in the study area (y axis in logarithmic scale) where stars indicate the five episodes with the highest number of hazard occurrences in the series.

**Methodology:** This section is currently difficult to follow, and the methods described need a bit more clarification and a justification of the methods choices. It is very important that everyone understands the event definition and selection.

P10 L 284: "the available precipitation data…" -> Why this assumption? With most of your datasets you could test this assumption.

*Also based on the comments received by the other reviewer, we realized that this part is not essential for the overall interpretation of methods and results, and it may reduce the readability of the text. We removed the two first assumptions and kept only the one about the use of daily precipitation for describing geohydrological hazard occurrences. We rephrased the sentence and added two references supporting the statement. In addition, we improved the motivation for the choice of the three metrics.*

*"The detection and characterization of past precipitation extremes in the study area are thus carried out by applying three different methodologies to daily precipitation fields over the 2003-2020 period. For this study, we assume that daily accumulated precipitation allows for a reasonable description of potential triggering conditions for the geohydrological hazards covered by the collected records (e.g., Leonarduzzi and Molnar, 2020; Schlögl et al., 2021). The*

*metrics adopted for event detection are chosen to consider three different aspects of extreme conditions, i.e., the spatial extent of intensities, the local intensity peak, and the combination of anomalies and their spatial extent."*

P11 L 296: "The ranked values ..." -> I would remove this sentence since you are not doing this.
*Done.*

P11 L 304: "is expected to capture ..." -> Doesn't this bare the risk of only sampling events that are located in the "high precipitation" areas?
*The comparison of this method with the other two allows for a better understanding of its ability to capture and describe extreme events. The metric does not bare the risk mentioned by the reviewer. The use of the local $99^{th}$ percentile serves only to identify extreme precipitation dates. Once precipitation events are selected, hazard records are matched with precipitation intensities on the corresponding extreme day and this match mostly depends on precipitation patterns and how well they are described by the different datasets over the domain, while the extreme metric is not used anymore. In any case, based on the results, local p99 is found to be comparable in describing the spatial match with hazard occurrences with other metrics, as also shown by the distribution of hazard records across precipitation classes (Table 3, reported in a subsequent answer below).*

P11 L314: "all wet-day values" -> Have wet days also been excluded from the other two methods? Or is it uncommon that the entire region has zero precipitation?
*It is important to clarify that the use of wet days regards only the calculation of the daily climatological means and standard deviations to apply for the standardization of daily records at each grid point. We adopted this definition in agreement with existing studies proposing the same metric (cited in the text). It does not affect the comparability of results with those obtained by the other two approaches. The $99^{th}$ percentile and the areal mean are computed spatially over grid cells in the grid for each day separately. Days with zero precipitation all over the domain can occur, and this would simply result in a null value for the metrics.*

P11 L317: "The product of the two..." -> Please be more explicit how you combined these two metrics.
*We rephrased by clarifying further how the magnitude is calculated.*

P11 L322-23: "To ensure that ... is retained." -> Does it make a difference whether you do the event filtering (i.e. clustering events within a 5-day window) before or after the ranking? When you remove the clustered dates from the ranked list, do you adjust the rank of the remaining list? Do you remove these dates entirely from all analysis?

*We used the term "ranking" to specify that we worked with sorted dates based on the specific metric considered. However, the actual rank assigned to the event is not particularly relevant, as we did not compare the event ranking in the list across metrics and datasets.*

*As regards the 5-day window selection, we improved the description of the selection procedure in Section 2.3.1. In the sorted list of dates, we decided to focus on the top 5 % portion as set of precipitation extremes to analyse, which corresponds to a total of 330 days. For each date, the corresponding event is represented by the 5-day window centred on that day. In case the central day of a 5-day window is included in another 5-day window, only the 5-day window centred on the highest-ranking day is kept as an event in the final 330 event set. It does not mean that dates are excluded completely, as they remain part of the 5-day window selected. When an overlapping event was discarded from the selection, we took another event from the sorted list until the target size of the event set (330) is reached.*

P11 L 324: "top 5% of sorted dates" -> Why the top 5%? And this 5% applies to the ranked list where dates belonging to the same event "within 5-days" have been removed?

*This was an arbitrary choice representing trade-off between having a sample of meaningful size and focusing on the most relevant events only. We tried to use 1 % of the dates, but the sample turned out to be too small to get robust results. As explained in the previous answer and in the manuscript, 5 % corresponds to 330 events in our case. To keep the target number of events fixed, any time one 5-day window is discarded, another one from the sorted list was added to the 5 % set. We made it clearer in the manuscript, and we motivated the choice of the top 5 % selection in Section 2.3.1, accordingly.*

P12 L 344-45: "A lower threshold ..." -> Did you also consider days with single hazards but clustered in time and space. Meaning that hazards from the same storm (say 3-days long) may trigger single hazards on each of these days in close spatial proximity. At the moment these hazard events would not be accounted for even though you might account for the 3-day precipitation event.

*To calculate the hit rate, we need a a-priori criterium to define first a well-defined hazard sample to use for quantifying the portion of it falling within a precipitation event (i.e., to use as denominator in the hit rate calculation). We decided to filter the hazard series by selecting only the dates with at least two hazard occurrences (called "hazardous dates") to reduce the "noise" of hazard records due to a substantial portion of dates in the time series with only one hazard occurrence over the study area (Figure 5c). Based on this definition, single hazards occurring on consecutive dates of a 5-day event window could not be included in the count used to estimate the hit rate. We remark that the hit rate analysis is intended to assess what portion of hazardous dates (defined as above) falls within extreme precipitation events represented by the top 5 % selection, and not vice versa.*

*The hazardous dates are considered for the hit rate analysis only (temporal consistency), while all hazard records within the 5-day windows of extreme events are used in the assessment of*

*spatial coherence. It means that also consecutive single hazard occurrences within the same precipitation episode enter the spatial analyses.*

*While revising the manuscript, we slightly modified the hit rate calculation. In the previous version, when multiple hazardous dates were found within the same 5-day window of a precipitation event, only one date was retained and the others removed from the count. However, we realized that it might unbalance the sample size of detected and undetected hazardous dates and lead to an underestimation of the actual hit rate as defined in the text (i.e., the portion of hazardous dates falling within an extreme precipitation episode as identified by each metric-dataset combination). In the revised version, all hazardous dates falling within a 5-day precipitation event are counted in the hit rate. The methodological description and results were updated accordingly (the revised hit rate table is also reported in a subsequent answer below).*

P12 L 356f: "Since ..." -> Create a new subsection. This will break up the methods description and it is easier to follow. This section would only cover the "spatial coherence" analysis. The previous one "hit and miss".

*We created another level of subsections splitting Section 2.3.2: the first paragraph described the temporal consistency analysis (hit rate) and the second one the spatial coherence analysis. We also slightly revised the text by remarking that the latter is intended to complement the hit rate assessment.*

P12 L 365: "To achieve this ..." -> On which basis are these percentile classes calculated? based on "all days" in the period 2003-2020, based on "wet days" in the period 2003-2020, or based on "extracted precipitation events" only?

*The percentiles are calculated for each extreme precipitation day separately based on the gridded daily precipitation values over the study domain (all grid points in the study area) on the date of the event. The percentiles are thus spatially defined and not temporally defined. We clarified the sentence in the methodology description. Please note that, following the suggestion from the other reviewer, we used the term "quantile" instead of "percentile".*

P12 L366: "Each hazard record ...." -> Is the "four nearest grid cells" applied irrespective of the spatial resolution? Isn't this likely penalizing the higher resolution? While the 1km grids suggest a higher accuracy, these datasets still have an effective resolution of 10-15km.

*Yes, it is applied for all products. We considered the four nearest cells to allow for a certain degree of uncertainty in the location of both hazard records and peak of precipitation intensity and to account for cases in which the hazard record is close to an adjacent grid cell in a higher precipitation class. The choice of four cells is arbitrary, and the reviewer is right to say that the resulting radius for searching the maximum value depends on the resolution of the product and might penalize the 1-km datasets. We rerun the analysis for all datasets by assigning to the hazard record the maximum precipitation class in a radius of 10 km. The 10-km radius,*

*which is consistent with the coarsest grid of ERA5-Land and the effective resolution expected for the high-resolution datasets, allows for a more robust search as it implies a different number of surrounding cells defined by the grid spacing. The results show more clearly that a higher portion of hazard records (more than 60 %) fall in the highest precipitation class for the events detected and described by the 1-km products, especially for INCA. The new analysis also highlights better the differences between the 1-km datasets and CERRA-Land, while the least pronounced distribution towards the highest classes is still confirmed for ERA5-Land. The same findings hold for all three extreme detection methods applied. We revised the methodology and the result sections, accordingly. The revised Table 3 is reported in a subsequent answer below.*

RESULTS

Have you also tested your results independently for the two hazard types (floods vs. mass movements)? I am wondering whether we can say something about whether the hit rate for floods is better than for mass movements. Both can be connected to intense precipitation, but I think it would be valuable if you could say floods are detected in x% of the cases and mass movements in y%. I am simply wondering whether the low hit rate (i.e. 50%) is due to the inability to match mass movement events which are very localized events compared to some of the flood events. For these events likely non of the datasets might be suitable.

*We calculated the hit rate for floods and mass movements separately both at annual and seasonal level. Resulting hit rates are not substantially different between the two categories, except for some higher hit rates for floods in the winter half-year. However, we found that it is difficult to derive some conclusive statements from this separate analysis and to estimate the contributing portion of individual hazard type to the hit rate obtained for the full hazard sample. In addition, flood records over 2003-2020 are significantly less (almost halved) than those of mass movements. We think that a type-specific analysis requires a more in-depth consideration of the different processes involved, time scales and precipitation triggering conditions, which goes beyond the scope of this study. For these reasons, we kept as main results the hit rate analysis based on the full set of hazard data, while the results for single hazard types have been included in Supplementary Material (Table S5 and Table S6) and mentioned in the Results to complement the main findings. Further details are reported in the next answer.*

*The hit rate tables for each analysis are reported also below:*

| ALL HAZARDS | Areal mean | | | Local p99 | | | Anomaly | | |
|---|---|---|---|---|---|---|---|---|---|
| | Year | Summer | Winter | Year | Summer | Winter | Year | Summer | Winter |
| **SPARTACUS-TST** | 50.6 | 49.4 | 52.9 | 55.1 | 57.2 | 50.7 | 53.8 | 55.8 | 50.0 |
| **INCA** | 52.1 | 51.7 | 52.9 | 56.0 | 57.6 | 52.9 | 54.1 | 56.5 | 49.3 |
| **CERRA-Land** | 48.9 | 48.7 | 49.3 | 52.3 | 53.9 | 49.3 | 49.1 | 48.3 | 50.7 |
| **ERA5-Land** | 46.7 | 43.1 | 53.7 | 49.6 | 48.7 | 51.5 | 44.7 | 41.6 | 50.7 |

**Table 2:** Hit rate (%) as the portion of hazardous dates (i.e., dates with at least two hazard records) occurring in coincidence of an extreme precipitation event (within a 5-day window centred on the extreme precipitation days) for each combination of

datasets and methods and by considering the whole year, summer (April to September) and winter (October to March) half years. The top 5 % most extreme precipitation dates over 2003-2020 are considered (330 events).

| FLOODS | Areal mean | | | Local p99 | | | Anomaly | | |
|---|---|---|---|---|---|---|---|---|---|
| | Year | Summer | Winter | Year | Summer | Winter | Year | Summer | Winter |
| SPARTACUS-TST | 48.7 | 44.6 | 60.5 | 58.0 | 58.0 | 57.9 | 58.0 | 57.1 | 60.5 |
| INCA | 51.3 | 49.1 | 57.9 | 59.3 | 57.1 | 65.8 | 56.7 | 55.4 | 60.5 |
| CERRA-Land | 48.7 | 43.8 | 63.2 | 56.0 | 54.5 | 60.5 | 49.3 | 44.6 | 63.2 |
| ERA5-Land | 42.7 | 35.7 | 63.2 | 47.3 | 42.9 | 60.5 | 42.7 | 36.6 | 60.5 |

**Table S5:** Hit rate (%) as the portion of hazardous dates (i.e., dates with at least two flood records) occurring in coincidence of an extreme precipitation event (within a 5-day window centred on the extreme precipitation days) for each combination of datasets and methods and by considering the whole year, summer (April to September) and winter (October to March) half years. The top 5 % most extreme precipitation dates over 2003-2020 are considered (330 events).

| MASS MOVEMENTS | Areal mean | | | Local p99 | | | Anomaly | | |
|---|---|---|---|---|---|---|---|---|---|
| | Year | Summer | Winter | Year | Summer | Winter | Year | Summer | Winter |
| SPARTACUS-TST | 51.8 | 51.0 | 53.1 | 55.4 | 57.2 | 52.2 | 53.1 | 54.6 | 50.4 |
| INCA | 54.1 | 53.6 | 54.9 | 55.4 | 56.7 | 53.1 | 53.4 | 55.7 | 49.6 |
| CERRA-Land | 50.2 | 51.5 | 47.8 | 53.7 | 55.2 | 51.3 | 50.5 | 50.5 | 50.4 |
| ERA5-Land | 48.2 | 44.3 | 54.9 | 50.5 | 49.5 | 52.2 | 45.9 | 42.3 | 52.2 |

**Table S6:** Hit rate (%) as the portion of hazardous dates (i.e., dates with at least two mass movement records) occurring in coincidence of an extreme precipitation event (within a 5-day window centred on the extreme precipitation days) for each combination of datasets and methods and by considering the whole year, summer (April to September) and winter (October to March) half years. The top 5 % most extreme precipitation dates over 2003-2020 are considered (330 events).

I think generally I would deemphasize the trend analysis. Detecting trends from the short time period and the shortcomings of the datasets inhibits any trustworthy trends.

*As suggested by the reviewer, we revised the Results section and added a first subsection (3.1) where we moved the comparison of climate datasets and the overview of hazard record distributions. The trend analysis was deemphasized, and the comparison of temporal variability is used to discuss the temporal consistency of the four datasets. We remarked in the text that, even though an increasing signal is apparent, the short length of the series does not allow for a robust evaluation of trends. We kept the trend assessment performed on the longer version of the datasets in the Supplementary Material as we used it to highlight that the increasing signal remains when extending the series in the past.*

*In this section, we have also distinguished the hazard records into the two classes (flood and mass movement) in the overview of monthly and annual distribution of hazard records (Figure 5).*

**Section 3.2.2:** I think the link between temporal hits (based on the areal statistics) and the link to the spatial hits can be strengthened. It would be interesting to know how many of the correctly detected hazard days (up to 50%) also show a correct spatial detection. Meaning that within your temporal search window and your spatial window you have an actual precipitation value which qualifies as an extreme value. You do analyze the connection already, but you could maybe make this a bit more implicit defining a spatial "hit rate".

*As also highlighted by the reviewer, we have already considered the spatial and temporal matching of precipitation events and hazard records. The spatial analyses (i.e., hazard locations across precipitation classes and precipitation intensity in the hazard proximity) have been already conducted considering the top 5 % precipitation events only, so that these percentages in Table 3 and the intensity distribution are already representative of the hazard days captured by the event selection.*

P13 L 387: "INCA is characterized by more pronounced increases in all statistics" -> This might be due to the inclusion of new radars in the recent years....
*We added that in the text.*

P14 L415: "72% to 88%" -> Are these numbers irrespective of the ranks, meaning that they agree on common unique days but can show inversed ranks? Did you also check the agreement in the rank locations of the events? Did you also quantify the agreement across the three event metrics?
*Yes, the numbers report the portion of common dates irrespective of their position in the top 5 % portion of the sorted lists of events. A quantification of the level of agreement in terms of event order across the dataset-method combinations is challenging and we think that the exact match may not be particularly relevant for the purpose of the study. However, we addressed this point by mentioning that only half of the 20 top-ranking events are in common in the selections based on different datasets or methods.*
*Based on the reviewer's question, we also assessed the agreement across the metrics and added this information to the text. The tables with explicit portions of overlaps across datasets and across metrics are now in Supplementary Material (Table S2 and Table S3).*

P14 L418-21: "For all methods ...." -> Where can we see this? How large is the overlap between the different datasets?
*As explained above, we added the tables with numbers in the Supplementary Material (Table S2 and Table S3). As explained in table captions, for each date in a certain set (dataset-method combination) of 330 events we checked whether it falls in any 5-day window centred on the events detected by another method-dataset combination.*

P16 L463: "correlation precipitation statistics and hazards ...." -> Are these correlations with all hazard days or with only hazard days with at least two reportings? I thought in your methods section you explained that you remove all days with only a single hazard reported.
*As described in Section 2.3.2, the filtering of hazard dates to retain only days with at least two hazard records was applied specifically to the hit rate calculation. For the correlation analysis, we first compared several extreme precipitation statistics with the complete time series (no filter applied) of hazard records to provide a preliminary assessment of the underlying relationship as described by the full set of available information. To complement it, we had already reported immediately after that by considering only dates with at least two hazard*

*records, the correlation increases. We made it clearer now in the methodology section that the filtering to hazardous dates was applied for performing the hit rate analysis only.*

Figure 8: Could you maybe add the statistics of the three metrics for this event and their ranks for each of the datasets.

*We added (now Figure 7) the event rank and several metrics for each dataset, specifically the spatial maximum and 99$^{th}$ percentile, the areal mean, the fraction of the domain where anomaly is above 2σ, the mean value of these anomalies and the resulting magnitude (i.e., mean value multiplied for the fraction of area).*

*Please note that these values refer to the single extreme date identified by each metric, while the underlying maps report the 5-day precipitation totals over 27$^{th}$ to 31$^{st}$ October 2018 as it represents the main documented window of the storm. It has been clarified in the figure caption.*

[Figure]

a) SPARTACUS-TST
Ranking - areal mean: 2; anomaly: 13; local p99: 4
Max: 184.6 mm; 99th percentile: 154.3 mm; areal mean: 62.2 mm
Fraction: 0.86; mean anomaly: 3.35; magnitude: 2.89

b) INCA
Ranking - areal mean: 3; anomaly: 17; local p99: 4
Max: 281.2 mm; 99th percentile: 166.5 mm; areal mean: 59.3 mm
Fraction: 0.73; mean anomaly: 3.51; magnitude: 2.56

c) CERRA-Land
Ranking - areal mean: 1; anomaly: 5; local p99: 4
Max: 147.2 mm ; 99th percentile: 120.4 mm; areal mean: 58.6 mm
Fraction: 0.83; mean anomaly: 3.47; magnitude: 2.88

d) ERA5-Land
Ranking - areal mean: 7; anomaly: 38; local p99: 6
Max: 84.0 mm; 99th percentile: 79.8 mm; areal mean: 44.7 mm
Fraction: 0.84; mean anomaly: 3.33; magnitude: 2.81

Precipitation totals [mm]
0   100   200   300   400   500   600

**Figure 7:** Cumulative precipitation during the Vaia storm, spanning from 27th to 31st October 2018, based on (a) SPARTACUS-TST, (b) INCA, (c) CERRA-Land, and (d) ERA5-Land. All hazard records in the study region that occurred in that 5-day interval are indicated by crosses. Metric values correspond to the single day of the identified Vaia event, which may vary by ± 2 days across different datasets and methods.

P18 L516-18: "Almost all datasets ..." -> Mention SPARTACUS-TST as an exception here. This dataset shows almost equal proportion of events in the second highest class (0.7-0.9), which represents more a moderate event intensity. My hypothesis would be that this is connected to the rather strict spatial rule of 4 closest grid cells and the inherent precipitation smoothing between stations. So, I think if you would extent the search radius to the scale of ERA5-Land (approx. 9 km) then the match to the highest precipitation class might be larger.

*We modified the method for assigning precipitation classes to the hazard records. We agree that considering the four closest grid cells for all datasets implies very different spatial scales*

*depending on the grid resolutions and might penalize too much the skills of km-scale products. We revised the methodology: instead of considering the four nearest cells, we searched for the maximum precipitation class within a 10-km radius around the hazard record in all cases. The 10-km radius accounts for the largest grid spacing of ERA5-Land as well as the effectively resolved scales of observation-based products. In this way, the portion of hazard records in the highest precipitation class for SPARTACUS-TST and INCA turns out to be remarkably greater than that in the [0.7-0.9) class. Moreover, it is now more evident that SPARTACUS-TST and INCA include a much larger portion of hazards in the highest class than reanalyses (Table 3 in the manuscript).*

| | | Quantile range | | | | | | |
| --- | --- | --- | --- | --- | --- | --- | --- | --- |
| | | [0-0.1) | [0.1-0.3) | [0.3-0.5) | [0.5-0.7) | [0.7-0.9) | [0.9-1] | Total |
| Areal mean | SPARTACUS-TST | 0.1% | 3.8% | 5.8% | 8.4% | 19.5% | 62.4% | 2,364 |
| | **INCA** | 0.3% | 2.6% | 4.2% | 8.5% | 17.6% | 66.7% | **2,390** |
| | CERRA-Land | 0.3% | 3.1% | 6.1% | 12.3% | 23.5% | 54.7% | 2,286 |
| | ERA5-Land | 2.3% | 5.9% | 11.2% | 23.5% | 21.8% | 35.3% | 2,239 |
| Local p99 | SPARTACUS-TST | 0.2% | 3.1% | 4.7% | 8.4% | 18.1% | 65.5% | 2,521 |
| | **INCA** | 0.6% | 2.1% | 4.0% | 7.2% | 15.5% | 70.5% | **2,692** |
| | CERRA-Land | 1.7% | 2.7% | 5.3% | 10.9% | 31.7% | 47.6% | 2,688 |
| | ERA5-Land | 1.8% | 6.7% | 12.8% | 22.4% | 23.3% | 32.9% | 2,325 |
| Anomaly | SPARTACUS-TST | 0.2% | 2.8% | 5.3% | 7.4% | 18.9% | 65.4% | 2,462 |
| | **INCA** | 0.7% | 2.1% | 3.8% | 8.0% | 16.4% | 69.1% | **2,460** |
| | CERRA-Land | 1.7% | 3.1% | 6.1% | 12.9% | 25.1% | 51.0% | 2,381 |
| | ERA5-Land | 2.1% | 6.0% | 11.1% | 24.2% | 21.3% | 35.4% | 2,176 |

**Table 3:** Distribution over different precipitation classes of hazards recorded in a 5-day window of the top 330 (5 %) events identified for each dataset-method combination. Precipitation classes are defined as quantile ranges of the gridded precipitation values over the study area. Values are reported as percentage of the total hazard records included in the 5-day windows of the top 5 % precipitation events (in the last column). For each method, the dataset reporting the highest total number of hazards included in the top 5 % precipitation events is in bold.

P19 L543: "more values in the upper tail" -> but also the lower tail. INCA shows a stronger left skewed distribution, which means that in INCA we have quite a few events with very low intensities.

*The left-skewed distribution in the original analysis is mainly a consequence of evaluating single INCA grid points (i.e., the closest grid point to the observed hazard). As discussed by Haiden et al. (2011), INCA precipitation at the grid-point scale is affected by limited spatial representativeness of rain gauges and residual radar uncertainties, so that small spatial displacement errors can result in unrealistically low intensities at individual grid points. In the revised analysis, we therefore consider a larger spatial neighbourhood (in a 10-km radius) and search for the maximum value over multiple grid points, which reduces these representativeness and displacement effects. As a result, the precipitation distribution becomes more symmetric and the previously pronounced left skewness mostly disappears (we reported below the revised figure). This behaviour is consistent with the recommendation implied in the INCA validation study that precipitation should be interpreted on spatially aggregated scales rather than at single grid points.*

[Figure]

**Figure 8:** Distribution of daily precipitation intensities in the spatial proximity of recorded hazards for all datasets based on the most intense 5 % precipitation events (330 events) detected by the local p99 method. The precipitation intensities are extracted from the closest grid point to each hazard location and the median values over all hazard records on the same date are displayed.

Figure 9: Can you change the line color of the median to black. The contrast is poor for the salmon color (INCA). You can also remove the unfilled outliers from the boxplots (non filled black circles), since these data points are shown in color anyway. Wouldn't it be better to simple show the precipitation intensities for all hazard events? As I understand you plot the distribution based on the p99 event selection with a hit in hazard, meaning that each distribution is based on a varying number of events. Ranging between 1693 to 2504 hazard events. Or am I interpreting this wrong?

*We revised the layout of the figure (now Figure 8, see previous answer) accordingly. As the distributions are based on the hazards falling in any 5-window of the 330 precipitation events, the reviewer is right to note that the number of hazards varies across datasets. As stated in Section 3.2.2, to reduce the disparities across datasets, the distributions are built by taking the median values of the intensities in the proximity of hazards occurring on the same date. In any case, the main shape of the distribution is expected to be mostly determined by the spatial scales resolved by the datasets and not by the specific size of the sample, which in any case is comparable among datasets and large enough (SPARTACUS-TST: 223 dates, INCA: 227 dates, CERRA-Land: 212 dates and ERA5-Land: 193 dates) to be considered representative of the dataset features.*

DISCUSSION

P21 L580-83: "The increasing trends of precipitation …" -> I would rephrase this sentence and focus on the temporal consistency rather than mentioning trends. The period is too short to

really say something about trends and in two out of the four datasets you have limitations in the consistency of the dataset.

*We agree with the reviewer's suggestion and we modified this paragraph accordingly (as also explained previously).*

P21 L584: "rise in the number of geohydrological hazards…" -> I think you need to mention here again the limitation of hazard record, which is likely strongly affected by reporting bias. I think you mentioned this is the methods section.

*We included the remark on the reporting bias in the text.*

P21 L603-04: "Although arbitrary, …" -> Have you considered to also filter by hazards in proximity, but one day apart?

*We did not fully understand the reviewer's point. The possible misalignment in time between hazard records and precipitation events is already taken into account by using the buffer of ± 2 days around each extreme precipitation date. In any case, we removed "although arbitrary" from the sentence as it might be misleading and does not add anything to the discussion.*

P22 L629-31: "The 5.5km grid and …" -> You can mention here Wood et al 2025 again to show that this is a consistent finding.

*Added.*

P22 L649-51: "The proportion of hazardous dates …" -> It might also suggest that the extreme event metrics are not capturing the essence of these events. "For instance, …." -> From at least one of your datasets (INCA) and potentially TST (if it is an hourly datasets) you could test this hypothesis. You could aggregate the hourly data by taking the daily maximum instead of the daily sums. Then do your analysis accordingly and check whether the detection rate is higher or lower.

*We modified the sentence by specifying that the low detection rate could also be due to the metric definition not able to fully describe the essence of events with hazard potential. As regards the use of sub-daily precipitation, since INCA is the only dataset offering sub-daily estimates, we cannot perform an intercomparison analysis, which remains the main focus of this work. Although the extension to sub-daily rainfall events is indeed a very relevant aspect to address, we prefer to include this evaluation in a future work, while mentioning it in the manuscript as a potential future study which can help with the interpretation of the temporal match of precipitation and hazard events.*

P23 L663-64: "This finding suggests…" -> However, as I mentioned before we can see for INCA many instances where precipitation in the vicinity of the hazard is very low (left skewed distribution).

*We revised the method to derive rainfall intensities in the hazard proximity by considering a 10-km radius as for the analysis of precipitation classes. This led to more stable and consistent results and reduce the left skewness of INCA (as explained previously).*

CONCLUSIONS

Shorten the conclusion to the most important take-aways that answer your research questions. You mention several details which you didn't really analyze and which are part of the methods and not a result of your comparison.

*We shortened the conclusions by removing details that were not part of our analyses and by making more prominent the key findings and messages.*

P23 L680: "mass waste" -> do you mean "mass movements"?

*Yes, corrected.*

P23 L681: "a significant increase in daily precipitation intensities" -> not really relevant and not a key finding of your study. I would remove this.

*Removed.*

P24 L696-98: "The study also showed ..." -> Not relevant, can be removed.

*Removed.*

P24 L700-05: "In future studies, ...." -> Move this to the discussion section

*Moved to the Discussion section.*

P24 L708: "meaningful information about ..." -> This statement contradicts a bit your finding of only 50% of hazards being detected by the extracted precipitation events.

*We revised this statement. We still remark that relatively simple statistics, when combined with sufficiently accurate precipitation products, can help the identification of extremes with hazard potential in a complex Alpine region. However, while the spatial coherence with hazard records of rainfall events from the transnational km-scale datasets already represents a solid basis for the characterization of precipitation intensities possibly triggering hazardous phenomena in the area, the hit rates slightly above 50 % are expected to increase if an in-depth evaluation on specific hazard processes is performed or more hazard-tailored extreme definitions are applied.*

---

## Author Comment (AC2)

**Detection and characterization of precipitation extremes and geohydrological hazards over a transboundary Alpine area based on different methods and climate datasets**

GENERAL COMMENT

Crespi, Enigl et al. study different rainfall datasets and test their potential for predicting geohazards. The test is conducted over a comparatively large are in parts of Italy and Austria. Results include how well the tested datasets and statistical descriptions of rainfall extremes identify storms and recommendations on how the which dataset should be used.

The strength I see in this study is more on the comparison of the datasets than on the testing of statistical thresholds to identify storms. Some of the datasets compared in this study are often used, also in other data-sparser regions of the world, making such a comparison useful. Which rainfall statistic is most powerful in predicting geohazards is a widely studied topic and I don't think the authors do this in much depth in this study. Furthermore, the results and conclusions are not presented in a very accessible way. I therefore mainly recommend streamlining and restructuring to frame the research in the right context and make it more accessible (i.e. higher impact). Nevertheless, I congratulate the authors on the work they've done so far, which I find useful and with practical impacts.

*We thank the reviewer for taking the time to read the manuscript and providing useful comments and suggestions. We need to clarify that the objective of our study is not to introduce novel statistics for describing triggering precipitation to use for hazard predictions. Instead, our aim is to systematically evaluate how well, through simple statistics or common definitions and through different datasets, we can identify precipitation events that are associated with the occurrence of hazards. Importantly, this assessment is carried out without imposing any assumptions on the physical or temporal dynamics of the hazard processes themselves, such as the role of antecedent conditions or specific triggering precipitation mechanisms.*

*Based on the reviewer's comment and the feedback provided by the other reviewer, the main changes applied to the manuscript are:*

- *We restructured Data and Methodology sections by moving part of the contents to the Results. In particular, the comparison among the datasets and the overview of collected hazard records are now in the new subsection 3.1 ("Precipitation statistics from meteorological datasets and hazard record overview").*

- *Based on the feedback received from the other reviewer, we performed the hit rate analysis for each hazard category separately (flood and mass movement), although results are used only for discussion purposes and we kept the hit rate based on the full hazard database (floods and mass movements together) as main analysis in the manuscript.*

- *We updated the hazard dataset by integrating the new version of WLV and GERIOS, which slightly increased the number of hazard records in our set. We updated all numbers and results based on the updated version.*

- *We revised Discussion and Conclusions by shortening them and making key messages more prominent and better related to the research questions of the study.*

*Specific comments are addressed below.*

I list my main comments below and line-by-line comments further down.

The paper cannot be read very fluently, and one often has to guess the intention of the authors with certain paragraphs/figures. For example, the methods around L180 on the rainfall datasets are a mix of methods, results and discussion. Likewise for the hazard catalogues, where trends are calculated and discussed in the methods (~L250). Also the discussion and conclusions could be better structured to better convey the key messages by adding subsections to the 3-page discussion that explicitly address the goals of the paper (testing the methods for extreme rainfall definition, testing different datasets, implications for practitioners).

*We thoroughly restructured the manuscript to better separate methods, results, and discussion. Specifically, we moved the rainfall dataset comparison and hazard catalogue overview to the new subsection 3.1 under Results. In this way, the methodological descriptions remain clearly distinguished from results and their interpretation.*

*In addition, we followed reviewer's suggestion and reorganized Discussion and Conclusions by explicitly address the main objectives of the study. In particular we split Discussion into four main paragraphs reflecting the structure of the analysis workflow (4.1 Temporal patterns of precipitation statistics and hazard records, 4.2 Methodological choices for comparing extreme precipitation events and hazard records, 4.3 Temporal match between extreme precipitation events and hazard occurrences, 4.4 Spatial coherence between extreme precipitation intensities and hazard records) and we shortened the Conclusions by focusing on key messages only.*

While I think the detection thresholds calculated from "areal mean", local p99" and "anomaly" I think generally are meaningful statistics to use. But I don't see much reasoning on why exactly these were chosen and there are not many references either in this part. Given that rainfall thresholds for geohazards has been a research topic for a long time, I miss the novelty compared to other studies or even just the justification for using exactly these statistics, while so many other statistics could be computed too (antecedent rainfall, multi-day cum. rainfall, …).

*It is important to clarify that the goal of our study is not to develop novel statistical methods for characterizing triggering precipitation for hazard prediction. Rather, our objective is to systematically evaluate how effectively simple statistics or widely used definitions for extreme characterization, applied across different datasets, can identify precipitation events associated with hazard occurrences. Crucially, this evaluation is performed without making any*

*assumptions about the physical or temporal dynamics of the hazard processes themselves, including the influence of antecedent conditions or specific precipitation-triggering mechanisms.*

*We chose these three statistics to measure the extremality in different features of rainfall: spatial extent (areal mean), local intensity (local p99) and magnitude, i.e., the combination of spatial extent and level of above-normal intensity (anomaly). We better specified the aim of the study in the Introduction:*

*"In this framework, the study aims to i) evaluate how metrics for precipitation intensity, not a-priori tailored to a specific hazardous process, enable to capture extreme events with triggering potential for geohydrological hazards over complex topography; ii) assess the suitability of precipitation datasets of different types and spatial resolution to describe extremes; iii) investigate the optimal combinations of metrics and datasets for characterizing extreme precipitation events and their spatio-temporal relation with hazard records. To answer these questions, three metrics measuring different aspects of rainfall extremes are calculated from 1-day precipitation fields of four meteorological datasets over a transboundary Alpine area between Italy and Austria and used to identify precipitation events over 2003-2020. Subsequently, they are compared with a harmonized archive of geohydrological hazard records to quantify the spatio-temporal match between identified events and observed records."*

*We also provided a motivation for the choice of the three metrics in the Methodology section: "The metrics adopted for event detection are chosen to consider three different aspects of extreme conditions, i.e., the spatial extent of intensities, the local intensity peak, and the combination of anomalies and their spatial extent."*

The dataset comparison is conducted by comparing the hit rate, eg in Table 2, at an artificially set threshold of top 5% rainfall events. However, from my experience it is more common and interesting to compare the predictive power of these datasets to separate hazardous from non-hazardous dates at a range of thresholds. For landslide early-warning, it is almost standard to report receiver operating characteristics. You will easily find references on this and the statistics can be calculated from the data you have with the eg the scikit learn library in python (eg ROC-curve).

*We initially computed receiver operating characteristic (ROC) metrics as part of our analysis. However, our case differs from the standard ROC application. Because our study focuses exclusively on the top 5% most extreme events—identified by applying the respective methodological approaches to each dataset—the vast majority of days within the study period (2003-2020) are classified as non-events. This leads to a large number of "misses," which biases the ROC curve and limits its interpretability in our context.*

*A more meaningful ROC-based evaluation would require considering all days in the period together with the full set of reported hazards, which would result in a dataset-independent*

*analysis. Such an approach, however, is beyond the scope of this study. Importantly, our objective is not to develop or evaluate an early warning system, but to assess the ability of three different statistics to identify extreme days that led to hazards when applied across different datasets.*

Specific comments:
L17: can you say more about the three definitions? Abstract readers will want to know the temporal scales of your analysis.
*We added the temporal scale, i.e., we specified that we used 1-day precipitation totals in our study, and we explicitly reported the extreme aspect measured by each metric instead of listing metric names.*

L21-24: Please specify in the abstract which data products you are testing. Now it only becomes clear that ERA5-Land is bad. But what is good? What do you mean by «high-resolution observation»?
*We have updated the abstract to specify which data products are being tested. It now clearly indicates not only that ERA5-Land performs poorly, but also which datasets (INCA and secondarily SPARTACUS-TST) show better performance.*

L79-80: also, the cited papers all seem hydro/flood related but not landslides
*We revised the Introduction by citing more studies linking precipitation and geohydrological hazards, covering both floods and mass movements (e.g., Peruccacci et al., 2017; Steger et al., 2023; Vaz et al., 2018; Araújo et al., 2022; Banfi and De Michele 2024). We also revised the previous paragraph about extreme definition by reporting more hazard-related studies for both floods and mass movements (e.g., Barton et al., 2022; Meyer et al., 2022).*

L113: a short intro to this section and the reasoning on how you chose the datasets would be helpful here. Also, a table with key facts about the different datasets would be very helpful
*We added an introductory paragraph in Section 2.2.1 and added a summary table with the main dataset features in the Supplementary Material (Table S1):*
*"Four climate datasets covering the study area are selected to assess their ability to detect and characterise extreme precipitation events over the transboundary region. The selection aims to evaluate precipitation fields from different types of products i.e., observation-based grids against reanalyses, and across different spatial resolutions. Two regional products are considered as km-scale datasets, one based purely on the interpolation of in situ observations and one incorporating multiple sources including observations and weather radar fields. The state-of-the-art European reanalysis CERRA-Land at 5.5 km and the global reanalysis ERA5-Land at 9 km are chosen to account for two widely used large-scale products and to evaluate to what extent their precipitation fields are comparable with those resolved by regional datasets. Each dataset is described in detail in the following, while key facts of each product*

*are summarized in Table S1. To enable the comparison, all analyses were based on the congruent period 2003 to 2020, while each product was used in its native spatial resolution."*

L181-196: These paragraphs are a mix of methods, results and discussion. Furthermore, I miss the link to your study. Could you say why showing these monthly means is important for your study on extremes?

*The monthly means were intended to provide a preliminary description of the dataset features, which in turn might be reflected also in the representation of extremes, e.g., spatial patterns, resolved scales, and seasonality. However, we recognized that showing a different statistic is more appropriate given the objectives of our study. We have restructured the relevant paragraphs and displayed the monthly 99th percentile both in Figure 2 and Figure 3 of the revised manuscript (we reported them below). The previous version of Figure 2 displaying the monthly means is now in the Supplementary Material (Figure S2) and used to support this preliminary dataset comparison.*

[Figure]

**Figure 2:** Monthly 99th percentile of daily precipitation calculated over all days in 2003-2020 and all grid points in the study area for the four gridded datasets considered.

[Figure]

a) Winter (Oct-Mar)  b) Summer (Apr-Sep)

SPARTACUS-TST  SPARTACUS-TST

INCA  INCA

CERRA-Land  CERRA-Land

ERA5-Land  ERA5-Land

0   15   30   45   60   75   90   105   120   135
Precipitation [mm]

**Figure 3:** a) Winter half year (October to March) and b) summer half year (April to September) 99th percentile of daily precipitation totals over 2003-2020 in the study area based on SPARTACUS-TST, INCA, CERRA-Land and ERA5-Land. Each dataset is shown in its native spatial resolution.

L201: Please specify how you define "gravitational mass movement" as this is a very broad term. Does it include rock glaciers? deep-seated landslides or only shallow? Debris flows? Rockfall?

*Gravitational mass movements comprise the following process types: shallow landslides, debris avalanches, debris and mud flows, and rotational and translational slides. Rockfall events are not included in the analysis. The detailed list of hazard processes considered is reported at the beginning of Section 2.2.2.*

L250: these lines again seem like results to me. Unless they were taken from other studies, but then a citation should be enough.

*These considerations are based on the preliminary assessment we performed on collected hazards from the different catalogues and their distribution over the analysed period. To make it clear that they are results of our analyses, we moved this part to the Results section in the subsection 3.1, together with the comparison of precipitation fields from different datasets.*

L283-287: Do you have evidence from other studies to support these assumptions?

*Based also on the comments received from the other reviewer, we revised this part of the methodology. We kept only the assumption related to the use of 1-day precipitation fields and removed the other points in this list as they are not essential for the interpretation of our results and might be misleading.*

L363: should be "quantile" instead of "percentile"

*We corrected that.*

Table 3: I'm having troubles understanding this table. I think it shows into which quantiles the 330 events fall. So each row should add up to 100%, which it doesn't, probably due to rounding. What are "intersected hazards"? I couldn't find a definition in the text and I don't get why this number differs among datasets.

*For each one of the 330 events, we classified the corresponding 1-day precipitation field into classes based on the quantile ranges of daily precipitation on that date over the domain. The quantile classes are thus relative to the spatial precipitation field of the extreme event. Then for each hazard recorded within the 5-day window of the precipitation event we extracted the precipitation class corresponding with the hazard location. By repeating this analysis over all 330 events, we got the total number of hazard records located in each class and we converted it into a percentage based on the total number of hazard records within all 330 events (last column in Table 3). The fact that the rows did not add up to 100% was a matter of rounding, we reported them with one decimal place in the revised version of the Table (see below). Please note that numbers in Table 3 have been updated based on the updated hazard dataset and the revised method for class assignment.*

*In the previous version we searched for the maximum precipitation class over the four nearest cells to the hazard record, which corresponds to a different searching radius depending on the resolution of the precipitation product and might penalize the 1-km datasets. We rerun the analysis for all datasets by assigning to the hazard record the maximum precipitation class in a radius of 10 km. The 10-km radius, which is consistent with the coarsest grid of ERA5-Land and the effective resolution expected for the high-resolution datasets, allows for a more robust search as it implies a different number of surrounding cells defined by the grid spacing of each product. The results show more clearly that a higher portion of hazard records (more than 60 %) fall in the highest precipitation class for the events detected and described by the 1-km products, especially for INCA.*

| | | Quantile range | | | | | | |
|---|---|---|---|---|---|---|---|---|
| | | [0-0.1) | [0.1-0.3) | [0.3-0.5) | [0.5-0.7) | [0.7-0.9) | [0.9-1] | Total |
| Areal mean | SPARTACUS-TST | 0.1% | 3.8% | 5.8% | 8.4% | 19.5% | 62.4% | 2,364 |
| | **INCA** | 0.3% | 2.6% | 4.2% | 8.5% | 17.6% | 66.7% | **2,390** |
| | CERRA-Land | 0.3% | 3.1% | 6.1% | 12.3% | 23.5% | 54.7% | 2,286 |
| | ERA5-Land | 2.3% | 5.9% | 11.2% | 23.5% | 21.8% | 35.3% | 2,239 |
| Local p99 | SPARTACUS-TST | 0.2% | 3.1% | 4.7% | 8.4% | 18.1% | 65.5% | 2,521 |
| | **INCA** | 0.6% | 2.1% | 4.0% | 7.2% | 15.5% | 70.5% | **2,692** |
| | CERRA-Land | 1.7% | 2.7% | 5.3% | 10.9% | 31.7% | 47.6% | 2,688 |

| | | | | | | | | |
|---|---|---|---|---|---|---|---|---|
| | ERA5-Land | 1.8% | 6.7% | 12.8% | 22.4% | 23.3% | 32.9% | 2,325 |
| Anomaly | SPARTACUS-TST | 0.2% | 2.8% | 5.3% | 7.4% | 18.9% | 65.4% | 2,462 |
| | **INCA** | 0.7% | 2.1% | 3.8% | 8.0% | 16.4% | 69.1% | **2,460** |
| | CERRA-Land | 1.7% | 3.1% | 6.1% | 12.9% | 25.1% | 51.0% | 2,381 |
| | ERA5-Land | 2.1% | 6.0% | 11.1% | 24.2% | 21.3% | 35.4% | 2,176 |

**Table 3:** Distribution over different precipitation classes of hazards recorded in a 5-day window of the top 330 (5 %) events identified for each dataset-method combination. Precipitation classes are defined as quantile ranges of the gridded precipitation values over the study area. Values are reported as percentage of the total hazard records included in the 5-day windows of the top 5 % precipitation events (in the last column). For each method, the dataset reporting the highest total number of hazards included in the top 5 % precipitation events is in bold.

L695: can you provide an example for an application requiring "accurate description of precip fields"?

*We added some examples in the text (hydrological modelling, early warning systems for floods and water-resource-related applications).*